# Firing patterns of ventral hippocampal neurons predict the exploration of anxiogenic locations

**Hugo Malagon-Vina[1]\*, Stéphane Ciocchi[1,2]\*[†], Thomas Klausberger[1]\*[†]**

[1]Division of Cognitive Neurobiology, Center for Brain Research, Medical University of Vienna, Vienna, Austria; [2]Laboratory of Systems Neuroscience, Department of Physiology, University of Bern, Bern, Switzerland

**Abstract** The ventral hippocampus (vH) plays a crucial role in anxiety-related behaviour and vH neurons increase their firing when animals explore anxiogenic environments. However, if and how such neuronal activity induces or restricts the exploration of an anxiogenic location remains unexplained. Here, we developed a novel behavioural paradigm to motivate rats to explore an anxiogenic area. Male rats ran along an elevated linear maze with protective sidewalls, which were subsequently removed in parts of the track to introduce an anxiogenic location. We recorded neuronal action potentials during task performance and found that vH neurons exhibited remapping of activity, overrepresenting anxiogenic locations. Direction-dependent firing was homogenised by the anxiogenic experience. We further showed that the activity of vH neurons predicted the extent of exploration of the anxiogenic location. Our data suggest that anxiety-related firing does not solely depend on the exploration of anxiogenic environments, but also on intentions to explore them.

**\*For correspondence:**
hugo.malagonvina@meduniwien.
ac.at (HM-V);
stephane.ciocchi@unibe.ch (SC);
thomas.klausberger@
meduniwien.ac.at (TK)

[†]These authors contributed equally to this work

## Editor's evaluation

This paper is expected to be of interest to systems neuroscientists in the fields of emotion, hippocampal function, and anxiety-related behavior. The authors performed recordings in the ventral hippocampus and show that (1) place fields become concentrated near the open areas of a maze, (2) direction-dependent coding decreases in these open areas, and (3) ventral hippocampal population activity in the closed area can be used to predict how mice explore the open area in the immediate future. These valuable findings provide convincing support for the potential role of the ventral hippocampus in the exploration of anxiety-provoking environments.

## Introduction

In the *Epistulae Morales ad Lucilium*, Seneca wrote: "There are more things, Lucilius, likely to frighten us than there are to crush us; we suffer more often in imagination than in reality". This sentence, from one of the key figures of the school of stoicism, describes inner fear within our imagination in the absence of a direct fear-provoking stimulus. Nowadays, even though anxiety and fear correspond to the same theoretical construct in some literature, anxiety differentiates from the latter based on the potential nature of the threat in the absence of an imminent harmful stimulus (*Calhoon and Tye, 2015*; *Davis et al., 2010*; *Steimer, 2002*). Anxiety disorders are becoming more commonly reported: 12-month prevalence estimates on mental disorders show that at least 14% of people in the European Union suffer from anxiety disorders (*Wittchen et al., 2011*), and around 31% of people in the United States have experienced some type of anxiety disorders in their lifetime (*Kessler et al., 1994*).

Different brain areas play a role in the underlying circuitry of anxiety (*Sandford et al., 2000*). Stimulations in the brainstem, more precisely in the periaqueductal grey matter or the locus coerulus are specifically involved in the symptomatology of anxiety (*Graeff et al., 1993*; *Redmond and Huang, 1979*). Some studies have also shown how the amygdala plays a role in humans suffering from anxiety disorders (*Birbaumer et al., 1998*; *Davidson et al., 1999*) or in animal models with generalised fear responses associated with anxiety (*Gründemann et al., 2019*; *Tovote et al., 2015*; *Wolff et al., 2014*). Also, by using the elevated plus maze (EPM) as an anxiety task (*Pellow et al., 1985*), Tye and colleagues induced anxiolytic effects by targeting projections from the basolateral amygdala (BA) to the central nucleus of the amygdala (*Tye et al., 2011*). Additional structures are involved in anxiety and include the bed nucleus of the stria terminalis, whose subdivisions are differentially involved in anxiety responses (*Duvarci et al., 2009*; *Kim et al., 2013*), and the medial prefrontal cortex (mPFC), which has also been directly linked to anxiety processing in both humans (*Rauch et al., 1997*) and rodents (*Park et al., 2016*; *Shah and Treit, 2003*).

Last but not least, the ventral hippocampus (vH) plays a critical role in anxiety behaviour. Lesions in the vH induce anxiolysis during the exploration of elevated open arenas (e.g. EPM) (*Jimenez et al., 2018*; *Kjelstrup et al., 2002*; *Padilla-Coreano et al., 2016*), or generally in tasks associated with approach-avoidance conflicts (*Schumacher et al., 2018*). Neurons recorded in the vH showed increased firing in locations with elevated anxiety (*Ciocchi et al., 2015*; *Jimenez et al., 2018*). Likewise, projections neurons from and to the vH exhibit anxiety-related activity: amygdala projections to the vH are specifically shaping anxiety-related behaviour during the exploration of an EPM (*Felix-Ortiz and Tye, 2014*; *Pi et al., 2020*). Similarly, the reciprocal connection (from vH to BA) is involved in the expression of context-dependent fear memories (*Kim and Cho, 2020*). Furthermore, information related to anxiety in the vH is directly routed to the mPFC (*Ciocchi et al., 2015*), and synchronised activity in this monosynaptically connected long-range circuit (*Adhikari et al., 2010*; *Adhikari et al., 2011*) is essential for the expression of anxiety behaviour. Motivated by the critical role of the dorsal hippocampus (dH) region in the encoding of spatial information (*O'Keefe, 1976*; *O'Keefe and Nadel, 2011*), several studies focusing on the vH have described its involvement in spatial coding (*Jung et al., 1994*; *Poucet et al., 1994*). Yet, probably due to its anatomical location and the difficulties to record action potentials of single-unit activity in the vH, the individual neuronal dynamics associated with the changes between anxiogenic and non-anxiogenic states during spatial navigation remain poorly understood.

To study the neuronal dynamics associated with anxiety behaviour in the vH, we simultaneously recorded the activity of individual neurons during the exploration of the EPM, as well as during the exploration of a novel behavioural paradigm, the elevated linear maze (ELM). During the same recording session, we modified the ELM from a non-anxiogenic to an anxiogenic configuration, while recording the activity of the same individual vH neurons. This enabled the investigation of the neuronal dynamics within the vH underlying different anxiety states. We specifically examined the remapping of neuronal activity at the single neuron and population level as animals transitioned from a non-anxiogenic configuration to an anxiogenic one. Collectively, the results of this study show that the neuronal activity in the vH does not simply reflect anxiogenic locations but that it is dynamically modulated by the experience and expectation of anxiety during spatial navigation.

## Results

### The firing activity of vH neurons is dynamically modulated during EPM exploration

Rats (n=6) freely explored an EPM consisting of two opposite arms with protective sidewalls (closed arms) and two opposite arms without sidewalls (open arms) (*Figure 1A*). The rats exhibited strong anxiety-related behaviour by spending most of the time in the closed arms, avoiding the more anxiogenic open arms and the centre (*Figure 1A*, bottom; closed vs open, p=9.5615e-10; closed vs centre, p=9.5657e-10; one-way ANOVA, Tukey-Kramer for multiple comparisons). While rats explored the EPM, we recorded neuronal activity in the vH with tetrodes (*Figure 1B* and *Figure 1—figure supplement 1A*) and isolated individual spikes from different single neurons (n=98). We identified vH neurons with previously described activity patterns (*Ciocchi et al., 2015*), exhibiting preferential firing in the open arms, closed arms, or centre of the EPM (*Figure 1C*). To understand the effect of the open areas

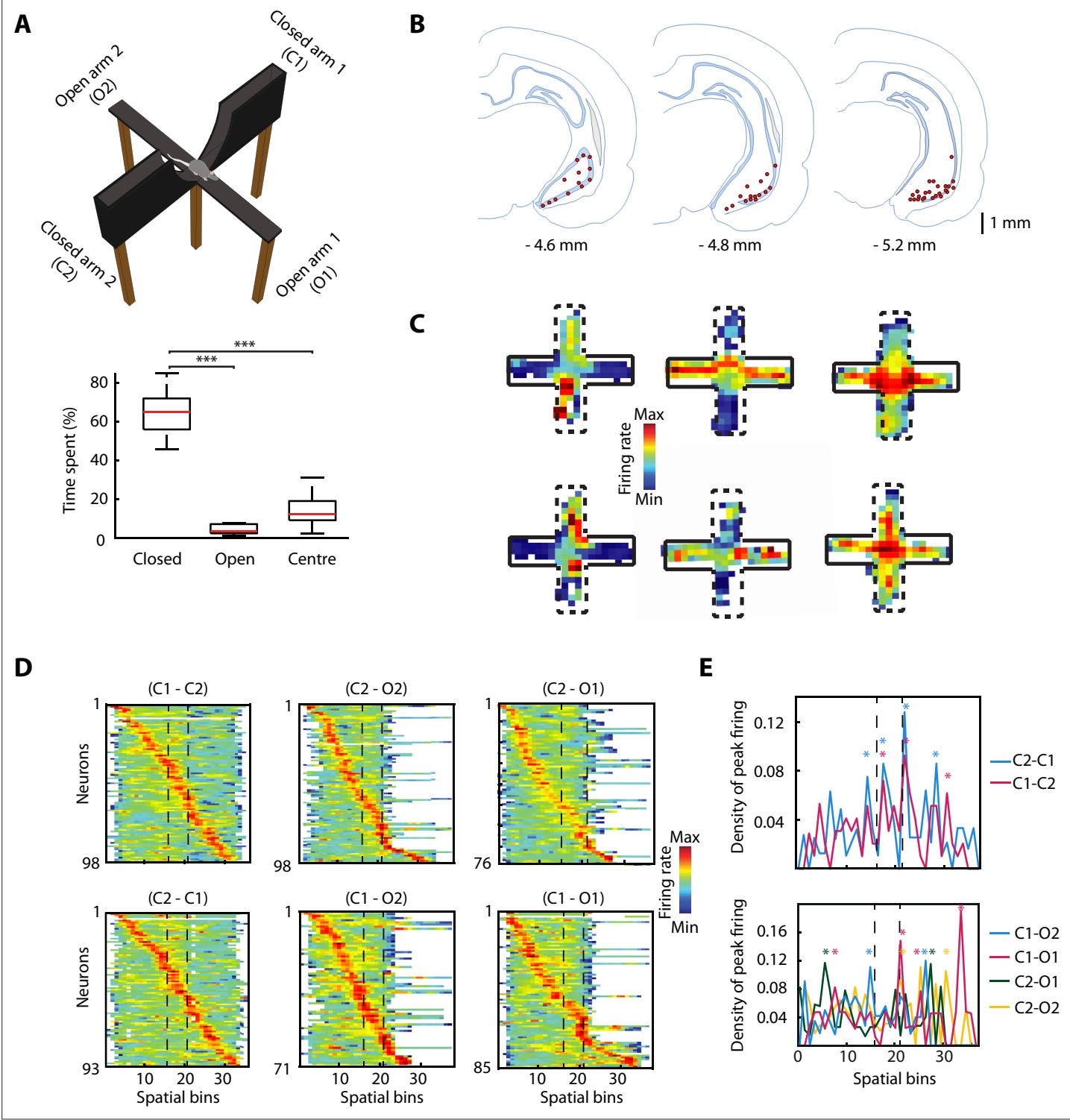

**Figure 1.** The activity of ventral hippocampal neurons is dynamically modulated during elevated plus maze (EPM) exploration. (**A**) Top, picture of the EPM. Bottom, the percentage of the time spent in different areas by the rats (n=6) during the exploration of the EPM. The time spent in closed arms is significantly higher than in the open arms and the centre (p=9.5615e-10, p=9.5657e-10, respectively. One-way ANOVA, Tukey-Kramer for multiple comparisons, n=16 sessions). (**B**) Location of tetrodes in the ventral hippocampus are indicated by red dots in three consecutive coronal sections (n = 47, number of rats = 8, one additional rat, for which histological location could not be confirmed, was included based on: insertion coordinates, oscillatory LFP profile, and similarity of neuronal activity). (**C**) Firing rate of six individual neurons during the exploration of the EPM. Full black lines denote a closed arm while the dotted lines indicate an open arm. Note three different anxiety-related activity patterns: increased firing in the open arms (left) or in the

*Figure 1 continued on next page*

*Figure 1 continued*

closed arms (centre) or in the centre (right). (**D**) Z-transformed firing rates (colour-coded) of ventral hippocampal neurons during the exploration of the EPM, separated by trajectories and sorted by the spatial location of their peak firing activity. Dotted lines indicated the centre area. (**E**) Top, density plot of the peak firing activity location for all neurons recorded during the journeys from a closed arm to the other during EPM exploration. Stars indicate the bins with significant higher density, calculated by bootstrapping (see Materials and methods). Note the increased number of peak activity at the centre (i.e. the only open area for these trajectories). Bottom, same as on the top, but for the trajectories between a closed arm and an open arm. The dotted lines denote the beginning and end of the centre area.

The online version of this article includes the following figure supplement(s) for figure 1:

**Figure supplement 1.** Nissl-stained sections and peak densities.

on the neuronal activity, we defined six possible trajectories taken by animals during EPM exploration (from one closed arm to any other arm). After linearising the trajectories (see Materials and methods), we organised the activity of the recorded neurons based on the spatial location of their peak (i.e. maximal) firing activity (*Figure 1D*). We observed that the peak firing activities of individual neurons spanned the entire maze even though the exploration of open arms was minimal. When plotting the peak firing density across the maze (normalised by the total number of neurons possibly firing in a spatial bin) the activity of vH neurons was concentrated around the centre of the maze when shuttling from one closed arm to the other one (*Figure 1E*, top, stars indicate bins with significant higher density; bootstrapping; see also *Figure 1—figure supplement 1B*). In some trajectories, a similar effect is observed when animals shuttled from one closed arm to an open one (*Figure 1E*, bottom, starts indicate bins with significant higher density; bootstrapping; see also *Figure 1—figure supplement 1B*). Even though the reduced number of entries to the open arms and exploration of open areas prevents a powerful statistic calculation, the observed effects imply that open areas are relevant for the neuronal activity of the vH.

However, these results suffer from important limitations: First, the extremely low and sporadic exploration of the open arms does not provide robust data sampling to test hypotheses regarding the neuronal computations associated with the exploration of an anxiogenic location. Second, the non-anxiogenic and the anxiogenic areas of the EPM are constantly present. This means that the characterisation of the anxiety states experienced by animals is not trivial as these may feel continuously anxious not solely in open spaces but also when considering to visit an open arm, while being in the closed arms of the EPM. In order to discriminate between neuronal activity related to spatial exploration from anxiety-related exploration and to motivate the exploration of anxiogenic areas for quantitative evaluation of the associated neuronal activity, we developed a novel behavioural paradigm to better control the transitions between anxiety states and the extent of exploration of anxiogenic areas.

## Removal of protective sidewalls along an ELM induces anxiety behaviour

We developed an ELM, which consisted of an elevated linear track with removable protective sidewalls. This allowed to have sidewalls either all along the entire ELM, called closed-closed (CC) configuration, or to have the sidewalls removed from half of the track, called closed-open (CO) configuration (*Figure 2A*). The rapid removal of the sidewalls enabled the alternation between a non-anxiogenic and an anxiogenic configuration within the same maze and in a single recording session. To control for neuronal activity-associated differences between two dissimilar areas in the linear maze exploration, while maintaining non-anxiogenic locations, we modified temporarily the visual appearance of sidewalls and floor texture in one half of the track, creating a new and enriched environment, without being anxiogenic. This configuration was termed closed-texture (CT) configuration (*Figure 2B*). Rats (n=6) were motivated to fully explore the ELM, by shuttling from one end of the track to the other one over numerous trials, to receive food rewards (*Figure 2C* and *Figure 2—figure supplement 1A, B*). To assess whether the removal of sidewalls induces behavioural readouts of anxiety similar to those of the EPM, we calculated the percentage of time spent on each of the arms for each configuration (*Figure 2D*). No differences were observed in the time spent by rats on the different arms during non-anxiogenic explorations (CC, CT). On the contrary, a significant difference was found in the configuration with sidewalls removed (CO exploration, p=1.45e-05, Wilcoxon signed-rank). The time spent in the centre area (11.5 cm around the middle of the track) was also longer during CO exploration

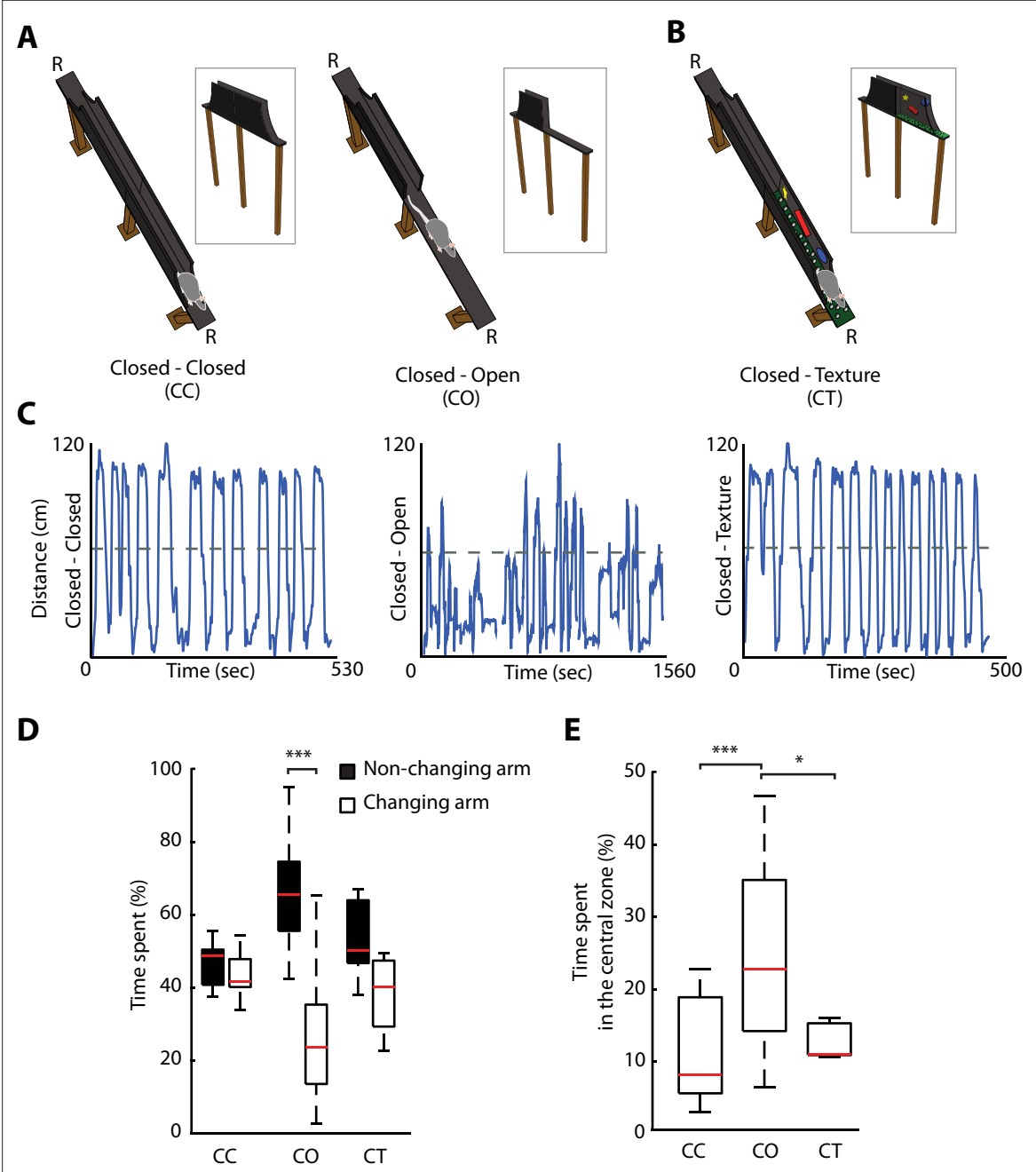

**Figure 2.** Removal of protective sidewalls along an elevated linear maze (ELM) induces anxiety behaviour. (**A**). ELM configurations with sidewalls along the entire track (closed-closed [CC], both arms closed, left), and with sidewalls removed for half of the track (closed-open [CO], called one arm closed – one arm open, right). Note the presence of a non-changing arm, while the other arm changes from a closed to an open configuration. R, indicates locations of food reward. (**B**) ELM configuration with both arms closed, but the floor texture and visual cues on inner-walls are changed in one of the arms (CT, closed and texture arm). (**C**) Linearised trajectories of a rat running in the three different ELM configurations during a single behavioural session. The grey line denotes the centre of the linear maze and the division between the two arms. (**D**) Time spent in both the non-changing and the changing arm in each configuration. Significant differences in the time spent appeared solely in the CO configuration (p=1.45e-05, Wilcoxon signed-rank, n=14 sessions). (**E**) More time spent in the central zone (defined as 11.5 cm around centre of the track) in the CO configuration in comparison to the CC and CT configurations (p=2e-05 and p=0.034, respectively, one-way ANOVA, Tukey-Kramer for multiple comparisons, n=14 sessions).

The online version of this article includes the following figure supplement(s) for figure 2:

**Figure supplement 1.** Analysis of runs along the elevated linear maze (ELM) for different configurations.

compared to CC or CT explorations (*Figure 2E*, p=2e-05 and p=0.034 against CC and CT, respectively, one-way ANOVA, Turkey-Kramer for multiple comparisons), suggesting of hesitations to enter this open area of the maze.

Overall, the removal of sidewalls along the ELM-induced anxiety behaviour that evolved during single behavioural sessions according to the anxiety content of each maze configuration.

## Overrepresentation and remapping of vH activity during anxiety

We recorded a total of 133 neurons with tetrodes in the vH (*Figure 1B* and *Figure 1—figure supplement 1A*), while the rats were exposed to the different configurations of the ELM. When the activity of individual neurons was sorted according to the spatial location of their peak firing activity, we observed that it spanned over the entire extent of the ELM for the three different configurations (*Figure 3A* and *Figure 3—figure supplement 1*). It is important to note that most of the neurons do not have a spatially restricted activity similar to a typical place cell in the dorsal CA1 hippocampus, but still they exhibit a peak of activity associated to a spatial bin (*Figure 3A* and *Figure 3—figure supplement 1C*). During the CO exploration, the distribution of peak firing activity was skewed towards the open area. We assessed the proportion of vH neurons with peak firing activity located on the different halves of the ELM during CC exploration. We found no differences in the proportion of peak firing activity located in each half of the CC configuration, even for the closed half that was going to be opened in the CO configuration (*Figure 3B*, left). Then, when one half was opened, a remapping of the neuronal activity was induced towards the anxiogenic location. We observed a higher proportion of peak firing activities located on the open arm (*Figure 3B* left, p=0.0121, chi-square test). We also compared the proportions of neurons that will 'Change to open' in contrast to the ones that will 'Change to closed'. 'Change to open' are neurons that, during CC configuration, had their peak firing located in the arm, which was to remain closed upon sidewall removal. After the removal, their peak firing activity was remapped to the open arm of the CO. 'Change to closed' are their counterpart neurons, meaning that during the CC configuration, their peak firing was located in the arm to be opened upon sidewall removal. After the removal, their peak firing activity was remapped to the closed arm of the CO. The difference in proportions between these two subpopulations was significant (*Figure 3B*, right, p=0.0066, chi-square test). *Figure 3C* shows the peak activity transition of every single recorded neuron before (CC) and after (CO) removing the sidewalls in half of the maze. Importantly, no differences were found for the location and location changes of peak firing activity during the exploration of the CT configuration using a novel texture and visual cues in the closed arm (*Figure 3D*), suggesting that dynamic remapping of vH activity is contingent to the experience of anxiety rather than to a stimulus-enriched environment or novelty.

Movement changes, mostly related to the running speed of the animal, could have been responsible for the neuronal activity changes observed during the exploration of the open arm. Speed-related modulation of hippocampal activity has been widely shown in the dorsal hippocampal region (*Czurkó et al., 1999*; *McNaughton et al., 1983*; *Wiener et al., 1989*). Using generalised linear models (GLMs), we aimed to capture the influence of the animal's instantaneous speed on the location-dependent activity for each cell (see Materials and methods). We found that the spiking activity of 49 neurons out of 133 (36.84%) was significantly modulated by the speed of the animal during the exploration of the CC configuration. Also, as expected, a higher number of neurons (72 out of 133, 54.14%) were significantly modulated by running speed during the exploration of the CO, in agreement with the speed changes related to the experience of an anxiogenic area. Next, we used the residuals of the GLM as an approximation of the neuronal spike-associated activity, corrected by the influence of the speed. Repeating the same analysis as in *Figure 3B and D* we found that a significant proportion of vH neurons changed the location of their peak firing activities from the closed area to the open area after sidewall removal (*Figure 3E*), in addition to no significant differences when exploring the CT configuration (*Figure 3F*). These results suggest that even though there is a modulation by the speed of the animal, there is also a prominent influence of the anxiogenic location in the neuronal activity irrespective of running speed.

As the population of cortical neurons is composed of pyramidal cells and interneurons, we aimed to sort the recorded neurons according to their putative identity by examining the spike shape and the firing rate of the individual neurons (*Csicsvari et al., 1998*; *Forro et al., 2022*; *Sirota et al., 2008*). Due to the stability of the spike shapes during the time of the recording (*Figure 3—figure*

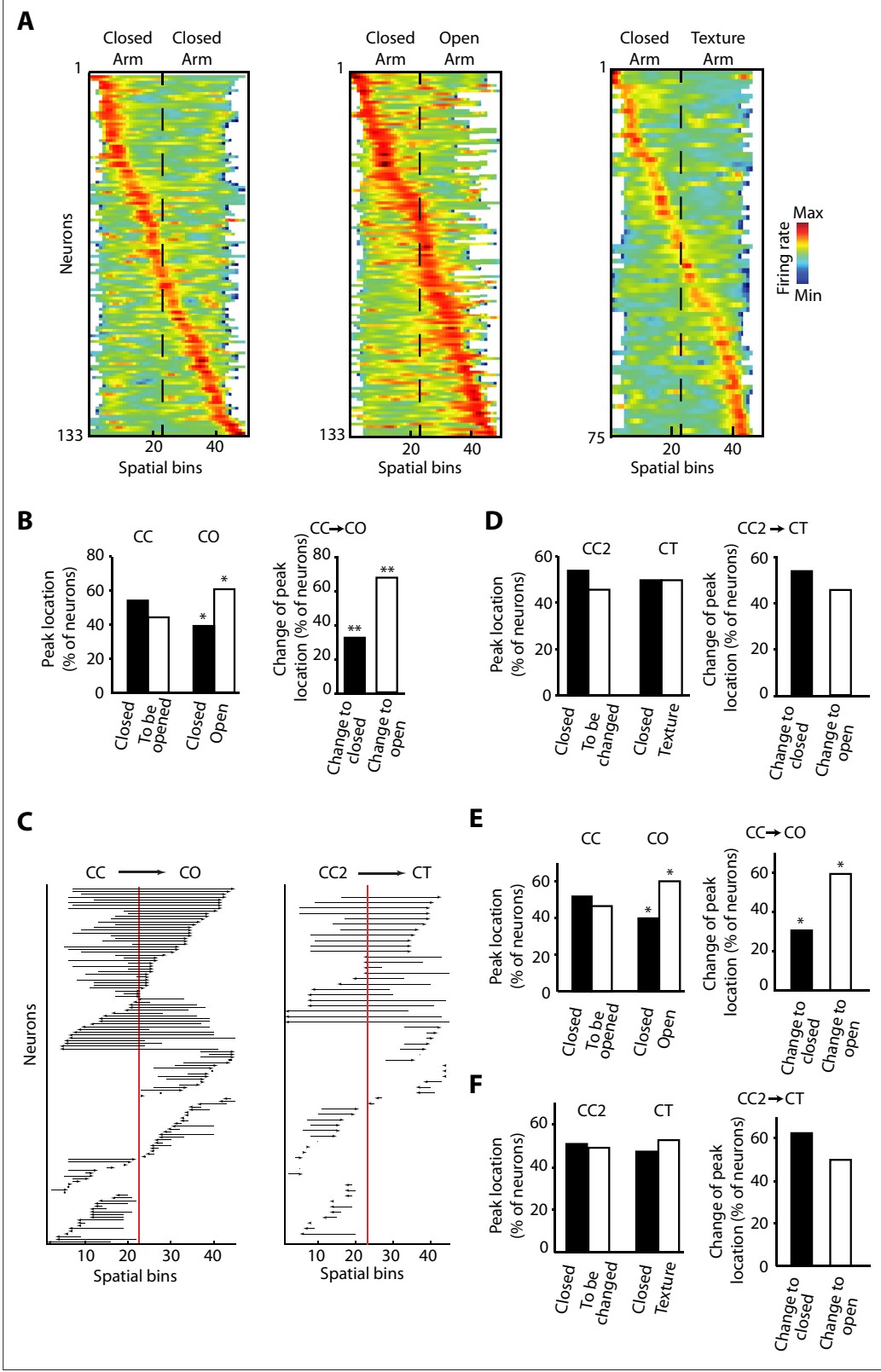

**Figure 3.** Overrepresentation and remapping of ventral hippocampal activity during anxiety. (**A**) Z-transformed firing rates (colour coded) of ventral hippocampal neurons during the exploration of the elevated linear maze (ELM) and sorted by the spatial location of their peak firing activity for the three configurations: closed-closed (CC) (left); closed-open (CO) (centre); closed-texture (CT) (right). The order of neurons is sorted for each configuration

*Figure 3 continued on next page*

*Figure 3 continued*

independently. Dotted lines indicated the centre of the maze. Note that firing rates of neurons remain mostly in the middle ranges (green colour), which is different from classical place cells of the dorsal hippocampus. There is an increased number of neurons with peak firing activity in the open arm. (**B**) Left, comparisons of the percentage of neurons with peak firing activity located in the different arms of the CC and CO configurations of ELM. Note the significant differences of the proportion of neurons with peak firing activity in the open arm (p=0.0119, chi-squared test, total n=133). Right, upon removal of the sidewalls, a larger proportion of neurons change the location of their peak firing activity from a previously closed to a currently opened arm (p=0.0066, chi-square test, total n=59). (**C**) Changes of the spatial location of the peak firing activity for individual neurons between different configurations. Each arrow denotes the remapping of the location of the peak firing activity between the CC (base of the arrow) and the CO configurations (arrowhead). The red line indicates the centre of the linear maze. Note that the peak firing activity of the neurons shifted towards the open arm when changing from the CC to CO configuration. (**D**) Same analysis as in (**B**) for the CT configuration. CC2 is a configuration with sidewalls along the entire track (fully closed) explored right before the presentation of the CT configuration. No significant differences were observed. (**E**) Left, comparisons of the percentage of neurons with peak firing activity (after correction of the speed influence in the spiking activity by using the residuals of the generalised linear model [GLM]) located in the different arms of the CC and CO configurations of ELM (p=0.019, chi-squared test, total n=133). Right, upon removal of the sidewalls, a larger proportion of neurons change the location of their peak firing activity (after correction of the speed influence in the spiking activity) from a previously closed to a currently opened arm (p=0.0189, chi-square test, total n=59). (**F**) Same analysis as in (**E**) for the CT configuration. CC2 is a configuration with sidewalls along the entire track (fully closed) explored right before the presentation of the CT configuration. No significant differences were found.

The online version of this article includes the following figure supplement(s) for figure 3:

**Figure supplement 1.** Neuronal activity during elevated linear maze (ELM) exploration.

**Figure supplement 2.** Spike-shape analyses.

**Figure supplement 3.** Remapping proportions for different subdivisions.

*supplement 2A, B*), we divided the recorded neurons in putative pyramidal cells (n=80, 60.15%) and putative interneurons (n=24, 18.05%) by using a threshold for the though-to-peak latency and the firing rate (put. pyramidal cells >0.3 ms and <20 Hz, respectively; and put. interneurons ≤0.3 ms and ≥10 Hz, respectively. *Figure 3—figure supplement 2B*, see Materials and methods). Twenty-nine neurons (21.80%) were not classified as either. Similar tendencies in the open arm-associated activity were observed for both, putative pyramidal cells and putative interneurons (*Figure 3—figure supplement 3A, B*).

Neuronal activity in the hippocampus is strongly affected by the emergence of sharp-wave ripples (SWR) (*Buzsáki et al., 1992*; *Csicsvari et al., 1999*; *Ylinen et al., 1995*). To control for this effect, we removed all SWR-associated spikes from the analysis, which resulted still in the same tendencies regarding open arm-associated activity (*Figure 3—figure supplement 3C*).

Collectively, these data indicate the dynamical recruitment of vH neurons when swapping between a non-anxiogenic (CC) to an anxiogenic configuration (CO), generating not solely a remapping of activity but also an overrepresentation of the anxiogenic area. As these effects were not detected during a novel but non-anxiogenic experience, we concluded that they might be attributed to anxiety processing during open-arm exploration rather than to a changed environment or novelty perception per se.

## Spatial properties of vH neurons affected by anxiogenic area

Observing the activity plots of vH neurons (*Figure 3A*), neuronal activity appeared to be spatially broader during CO exploration, suggesting that the anxiogenic area in the CO configuration selectively affected the spatial features of vH neurons. To investigate how CO exploration changed different spatial properties of neuronal activity, we calculated each neuron's spatial information, sparsity and coherence for each ELM configuration (*Skaggs et al., 1996*; *Figure 4—figure supplement 1A*, see Materials and methods). Specifically, we compared these three properties for neurons with peak firing rate located in the open arm of the CO configuration; we called these 'peak in open' (PiO) neurons. Reciprocally, neurons whose peak activity was located in the closed part of the CO configuration, were called 'peak in closed' (PiC) neurons. Interestingly, both PiO and PiC neurons showed similar properties: their spatial information diminished when going from a CC to a CO configuration (p=1.66e-05 for

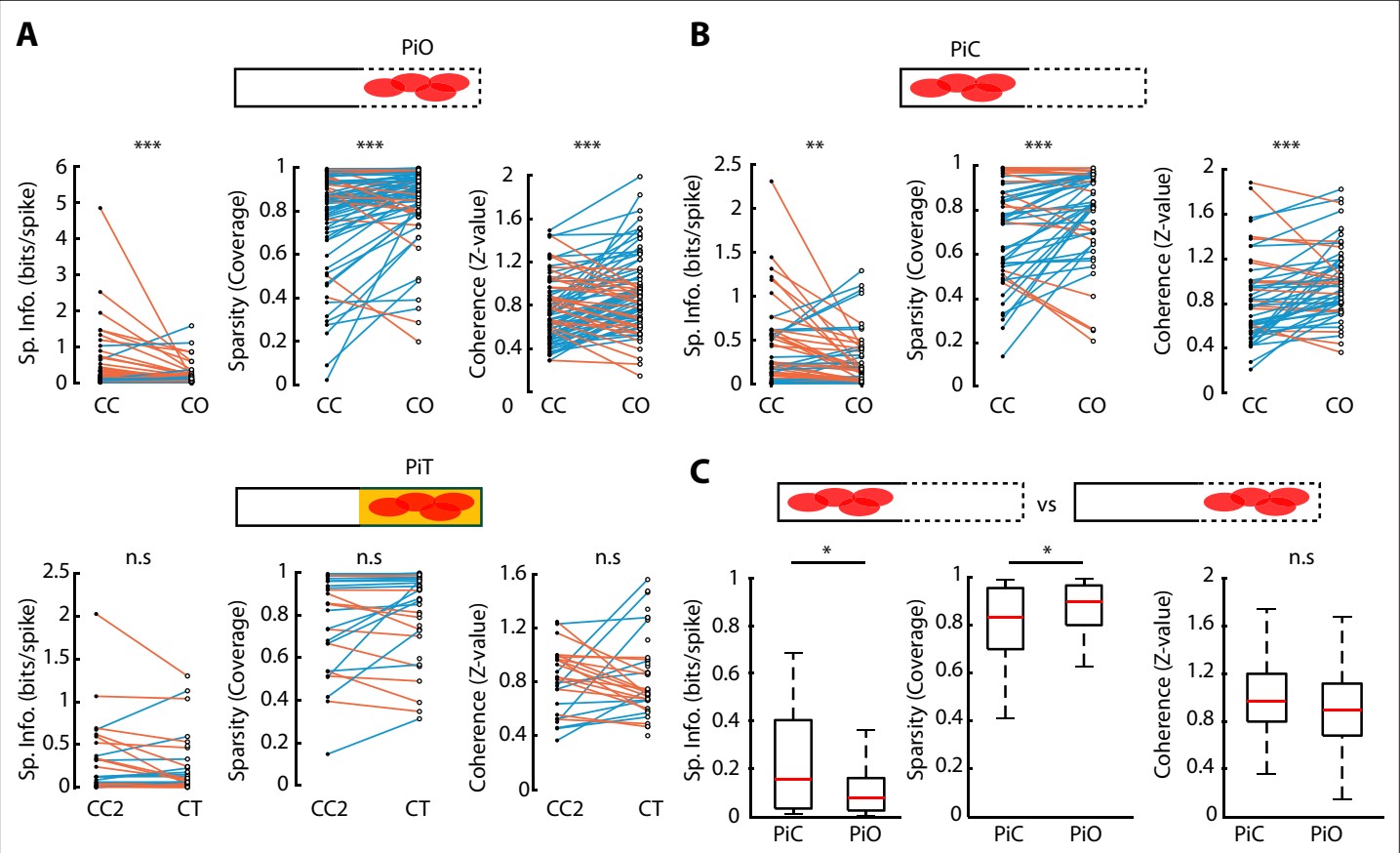

**Figure 4.** Spatial properties of neuronal activity during elevated linear maze (ELM) exploration. (**A**) Top, paired comparison of the spatial properties for PiO (firing peak in open area of the closed-open [CO] configuration) neurons between their activity in the closed-closed (CC) and the CO configurations. Spatial information (p=1.66e-05, Wilcoxon signed-rank test, n=81), sparsity (p=6.86e-05, Wilcoxon signed-rank, n=81), and coherence (p=1.88e-05, Wilcoxon signed-rank, n=81) were significantly different. Blue lines indicate an increase in the score while orange lines indicate a decrease. Bottom, paired comparison of the spatial properties for PiT (firing peak in the texture area of the closed-texture [CT] configuration) neurons between their activity in the CC2 and the CT configurations. No significant difference in spatial information, sparsity, or coherence was observed (n=38). (**B**) Paired comparison of the spatial properties for PiC (firing peak in closed area of the CO configuration) neurons between their activity in the CC and the CO configurations. Spatial information (p=0.0074, Wilcoxon signed-rank test, n=52), sparsity (p=9.68e-04, Wilcoxon signed-rank test, n=52), and coherence (p=2.86e-04, Wilcoxon signed-rank test, n=52) were significantly different. (**C**) Comparison of spatial information, sparsity, and coherence during CO configuration between PiC and PiO neurons. Significant differences are observed in spatial information (p=0.0130, Wilcoxon rank-sum test) and sparsity (p=0.0311, Wilcoxon rank-sum test).

The online version of this article includes the following figure supplement(s) for figure 4:

**Figure supplement 1.** Spatial and firing – properties of recorded neurons.

PiO and p=0.0074 for PiC, Wilcoxon signed-rank test), while their sparsity and coherence increased (p=6.86e-05 and p=1.88e-05 respectively for PiO; p=9.68e-04 and p=2.86e-04 for PiC, Wilcoxon signed-rank test). Such effects were not observed for neurons during changes from CC2 to CT configurations (*Figure 4A and B*). However, PiO spatial properties tended to be diminished compared to PiC neurons for both spatial information (p=0.0130, Wilcoxon rank-sum test) and sparsity (p=0.0311, Wilcoxon rank-sum test) (*Figure 4C*). Concerning neurons that remapped from the closed arm in CC to the opened arm in the CO configuration, these neurons showed significantly lower spatial information and significantly higher sparsity (p=0.0029 and p=0.0016, respectively, Wilcoxon rank-sum test) already in the CC configuration, prior to the CO configuration, with respect to the other neurons that did not remap (*Figure 4—figure supplement 1B*).

In addition to the spatial properties and general activity remapping, rate remapping represents another factor which indicates modulation by a new environment or stimulus (*Kaefer et al., 2019*; *Latuske et al., 2017*; *Leutgeb et al., 2005*). Even though the overall firing rates were not significantly

different between different configurations (*Figure 4—figure supplement 1C*), we calculated the remapping score (see Materials and methods) for each neuron between the CC and CO configuration and between the CC2 and CT configuration. The CC2-CT remapping score was significantly higher than the CC-CO (p=0.0358, Wilcoxon rank-sum test, *Figure 4—figure supplement 1D*).

Altogether, the results are in line with the experience of an anxiogenic environment (CO) and a non-anxiogenic but new environment (CT) being perceived. This seems to be supported by the observation that the spatial representation of all neurons responding to the anxiogenic area in the CO configuration was reduced (lower spatial information and higher sparsity) upon the removal of the walls, with this effect more pronounced for those neurons whose activity reached their maximum in the open area of the anxiogenic environment.

## The direction-dependent activity of vH neurons becomes homogenised following the introduction of an anxiogenic location

Hippocampal place cells have been reported to exhibit direction-specific spatial modulation of activity as animals run along a linear maze (*McNaughton et al., 1983*; *Muller et al., 1994*; *Royer et al., 2010*). This raises the question as to whether the activity of vH neurons, recruited by areas with increased anxiety content, are also modulated by the direction of the journey along a linear track. When monitoring the activity of individual vH neurons recorded during the exploration of the CC configuration (*Figure 5A*), we expectedly observed a profound direction-dependent difference. However, this direction-dependent neuronal firing of the same vH neurons became homogenised, meaning that it was very similar in both directions, once the sidewalls were removed and the rats were exploring the CO configuration. To quantify this phenomenon, we use the Spearman correlation as a measurement of place field similarity (PFS) of a neuron between its activity while the animal moved in one direction vs the other direction. The PFS value obtained from the correlation will grant an accurate approximation of the similarity between the neuronal activity of both directions. As expected, the PFS index of animals exploring the CC configuration was significantly different from the PFS index of animals exploring the CO configuration (*Figure 5B*, left, p=7.53e-09, Wilcoxon rank-sum test). Based on the values of the PFS indices (median $PFS_{CC}$ = –0.0539, median $PFS_{CO}$ = 0.5091), we inferred that the significant difference between the PFS distributions is due to a similar neuronal activity in both directions during the exploration of the CO configuration.

To corroborate that such a phenomenon was not due to the novelty of the arm, we calculated the PFS indexes of the neuronal activity during the exploration of a novel arm (CT), instead of an anxiogenic location (i.e. CO configuration). The PFS indexes between the exploration of the CC configuration prior to the CT (called CC2) and the PFS indexes during the CT configuration were not significantly different (*Figure 5B*, right, median $PFS_{CC2}$=–0.1279, median $PFS_{CT}$ = 0.0922). In general, all the PFS values during the exploration of CC, CC2, and CT were significantly lower than during the exploration of the CO configuration, as seen in the cumulative distribution plots of the PFS (*Figure 5C*, CO vs CC, p=7.53e-09; vs CT, p=4.6e-05; and vs CC2, p=1.68e-07; Wilcoxon rank-sum test). The same tendencies were seen after obtaining the neuronal spike-associated activity controlled by the speed of the animal (*Figure 5D and E*), as described previously and in the Materials and methods, as well as for putative pyramidal cells, putative interneurons, and, in general, neuronal activity after the removal of SWR-associated activity (*Figure 5—figure supplement 1A, B, C*).

In conclusion, even though there was direction-dependent neuronal activity in the vH during the exploration of a non-anxiogenic linear maze, this spatial dependency was reduced when animals encountered an anxiogenic location, and the spiking activity of the neurons tended to be homogenised independently of the direction of the animal.

## The activity of vH neurons predicts the extent of exploration of an anxiogenic location

We have shown that neurons of the vH change their activity patterns during the experience of anxiety. The described changes on neuronal activity were mainly related to the exploration of the open area. Next, we asked if neuronal activity during the exploration of the closed arm during a CO configuration was influenced by the upcoming anxiety states. Observation of behavioural readouts (*Figure 2*) seemed to indicate a hesitation to enter the opened arm in the CO configuration, and after entering, there not always a full commitment to explore it to the end. Therefore, we asked whether neuronal

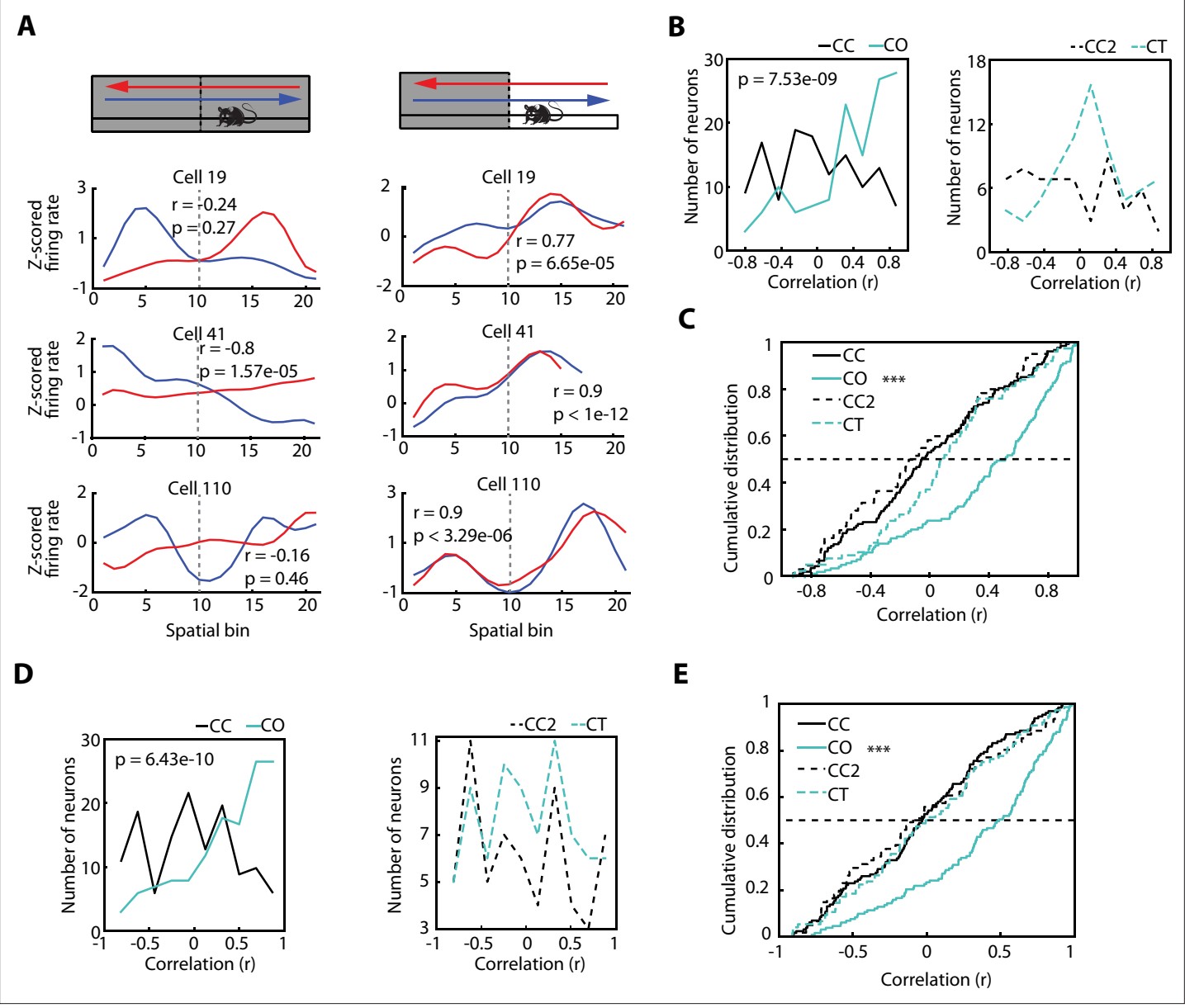

**Figure 5.** The direction-dependent activity of vCA1 neurons is homogenised after exposure to an anxiogenic location. (**A**) Neuronal activity of individual ventral hippocampus (vH) neurons while animals explored the elevated linear maze (ELM) in both the closed-closed (CC) and the closed-open (CO) configuration. Blue lines denote when animals headed towards the arm that will be open (in the case of the CC configuration) or is open (in the case of the CO configuration). Red lines denote when animals returned from this arm. Correlation values (Spearman correlation) indicate the similarity between the neuronal activities of both trajectories. (**B**) Histograms of the firing rate maps similarity index (place field similarity [PFS], see Materials and methods) for the activities of single neurons calculated in the two possible directions on the ELM (left to right and right to left). Left, PFS index is higher during the CO configuration (cyan) compared to the CC (black) configuration (p=7.53e-09, Wilcoxon rank-sum test, CC: n=128 and CO: 133). Right, PFS index is not significantly different during the closed-texture (CT) configuration (dotted cyan) compared to the CC2 (dotted black) configuration (CC2: n=60 and CT: n=75). (**C**) Cumulative distributions of the PFS indexes for the CC, CO, CT, and CC2 configurations. Note that during the CO configuration the PFS index of neurons is significantly higher compared to the other configurations (vs CC, p=7.53e-09; vs CT, p=4.6e-05; and vs CC2 p=1.68e-07; Wilcoxon rank-sum test). (**D**) Histograms of the firing rate maps similarity index (PFS, see Materials and methods) for the activities of single neurons (after correction of the speed influence in the spiking activity) calculated in the two possible directions on the ELM (left to right and right to left). Left, PFS index is higher during the CO configuration (cyan) compared to the CC (black) configuration (p=6.43e-10, Wilcoxon rank-sum test). Right, PFS index is not significantly different during the CT configuration (dotted cyan) compared to the CC2 (dotted black) configuration. (**E**) Cumulative distribution of the PFS indexes in (D) for the CC, CO, CT, and CC2 configurations. Note that during the CO configuration the PFS index of neurons is significantly higher compared to the other configurations (vs CC, p=6.43e-10; vs CT, p=3.06e-06; and vs CC2 p=4.32e-06; Wilcoxon rank-sum test).

The online version of this article includes the following figure supplement(s) for figure 5:

**Figure supplement 1.** Additional analysis on the direction-dependent activity of ventral hippocampus (vH) neurons.

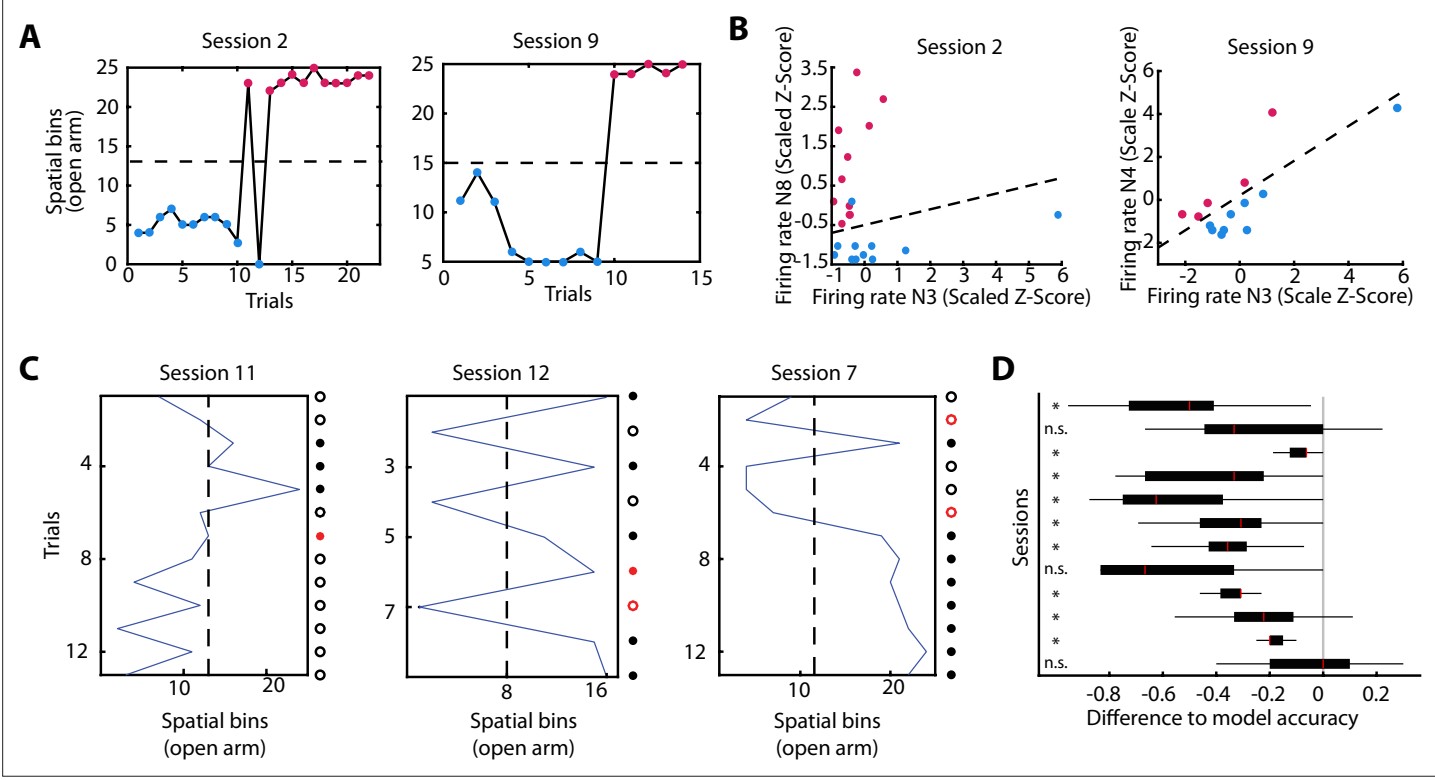

**Figure 6.** The activity of ventral hippocampal neurons predicts the extent of exploration of an anxiogenic location. (**A**) For two behavioural sessions, the furthest spatial bin visited on each trial is plotted. The dotted line indicates the spatial bin set as a criterion to define two different types of trajectories: proximal (blue dots) and distal (magenta dots) exploration. (**B**) For the two behavioural sessions in (**A**), the single-unit activity of two co-recorded neurons during the run on the closed arm are shown. The dotted line in the panel indicates the result of support vector machine (SVM) plane for the separation of both, distal and proximal, categories (see Materials and methods). Firing rate is given as the z-score scaled by the SVM (see Materials and methods). (**C**) Predictions of the SVM for three additional individual sessions. The blue line shows the furthest spatial bin reached during specific trials. The dotted line indicates the middle of the exploration of the open arm and divides the trials into proximal (left) and distal (right) explorations (used for the SVM classifier). Dots at the right of each plot show the trial by trial accuracy of the SVM classifier (see Materials and methods). Full dots show trials with distal explorations while circles show trials with proximal explorations. Red colour implies inaccurate prediction of the SVM classifier for that specific trial. (**D**) Normalised distributions of the SVM classifier performance (observed data – shuffled data) for each of the 12 sessions used. For each session, the neuronal pair with the best SVM performance was subtracted from the performance of the same pair with shuffled distal and proximal explorations (1000 shuffles). Therefore, the grey line at 0 marks the SVM classifier performance on the observed data and the boxplots show the differential performance of each of the shuffled iterations. Not significant (n.s.); significant (*, lower than 0.05). Boxplots show median (red line), 25th and 75th percentile.

The online version of this article includes the following figure supplement(s) for figure 6:

**Figure supplement 1.** Speed comparison and neuronal activity of individual cells.

**Figure supplement 2.** Further analysis on support vector machine (SVM) predictions.

firing in the closed area might be predictive of the extent of an upcoming exploration in the anxiogenic area. To do so, we divided spatial explorations into two groups depending on how far animals explored the open arm during a particular trial: proximal and distal exploration trials were defined using an individual threshold for each session ($Threshold = \frac{Furthest\ spatial\ bin - Nearest\ spatial\ bin}{2}$) (**Figure 6A**). We then tested if the neuronal activity before entering the open arm was informative of how far animals would venture into the open arm (proximal or distal explorations). We trained a support vector machine (SVM) using the neuronal activity for each possible neuronal pair in the closed arms and each recording day (see Materials and methods). Using a wrapper-like method, we then selected the neuronal pair with the highest performance on correctly predicting the extent of exploration for single trials on a given recording day (**Figure 6B** and **Figure 6—figure supplement 1A** for raster plots). Since the distribution of distal and proximal trials was not equal (50%–50%), we then determined if the performance of the optimal pair in a given recording day was significant by shuffling the trial IDs 1000 times and re-calculating the performance on a newly trained SVM. A recording day

was labelled as successfully predicted if the performance of the optimal neuronal pair was higher than the 95% percentile of the shuffled distribution. Classification performance for three sessions are shown in *Figure 6C*, where the red coloured dots imply an incorrect classification. The summary of the performance of the classifiers for all sessions is shown in *Figure 6D*, where the boxplots represent the performance of each of the 1000 shuffles but with the optimal performance of the session subtracted (grey line at 0 denotes the optimal performance and the boxplots the 'differential to the optimal' shuffled ones). We found that during individual recording sessions, the performance of the SVM was above chance levels (95th percentile) in 9 out of 12 sessions. Also, the performance of any pair was higher than the performance of the best pair's shuffled data, with few expected exceptions (*Figure 6—figure supplement 2A*). Because the area before entering the open arm of the CO configuration seemed to be a behaviourally important location, it might be possible that we were capturing the speed-related neuronal activity (due to a speed difference between trials in which the rat would visit the end of the arm, or would return early). However, we observed that the animals' speed before entering the open arm was not different between proximal or distal run endpoints, except for one session (*Figure 6—figure supplement 1B*). In addition, correct prediction of several sessions using the SVM classifier was achieved for controls in which speed or SWR-associated activity was corrected and also when restricting the pairs to be either only putative pyramidal cell pairs, or putative interneuron pairs (*Figure 6—figure supplement 2B*).

These analyses indicate that the neuronal representation of the anxiogenic location was not only dynamically modulated by the momentary experience of anxiety, but already existed in the closed arm, possibly reflecting the intention to venture into the open arm. This implies that neuronal activity within the vH can predict upcoming anxiogenic situations, even when animals are still located in a safer environment without a direct exposure to an anxiety-inducing location.

## Discussion

To investigate the neuronal dynamics governing anxiety behaviour in the vH, we recorded the activity of individual neurons while rats explored anxiogenic locations. In addition to the classical EPM, we used a novel ELM, which allowed us to rapidly change the anxiety content of the maze to expose rats to non-anxiogenic or anxiogenic configurations. We found that the neuronal activity of the vH exhibited a uniform spatial representation in the non-anxiogenic configuration of the ELM and that vH neurons displayed direction-dependent spatial firing when shuttling from one end of the ELM to another. When the anxiogenic location was introduced by removing the protective sidewalls from half of the track, the peak firing activity remapped towards the newly introduced anxiogenic location, the spatial properties of the neuronal activity diminished, and direction-dependent firing was homogenised. Of important note, neuronal activity in the closed arm of the ELM predicted the extent of the upcoming exploration in the open arm even before rats entered into the open anxiogenic location.

Much of the anxiety research in freely moving rodents has been relying on the EPM. Using the EPM, it has been shown that: amygdala projections to the vH control the expression of anxiety (*Felix-Ortiz and Tye, 2014*; *Pi et al., 2020*); there is an anxiety-associated neuronal activity in the vH routed to the mPFC *Ciocchi et al., 2015*; and that the vH-prefrontal pathway is critical for anxiety behaviour (*Adhikari et al., 2010*; *Adhikari et al., 2011*; *Ciocchi et al., 2015*). The first part of this manuscript focuses on the activity of neurons recorded in the vH during the exploration of an EPM. We divided the exploration of the EPM into different trajectories. We observed a localised increase in the density of peak firing activity when rats crossed the centre in all the different C-C or C-O trajectories (*Figure 1E*). This might be explained by the fact that not only the open arms of an EPM are anxiogenic, but also the centre zone (*Mendes-Gomes et al., 2011*). Unfortunately, it proved difficult to analyse these effects quantitatively, because rats explored the open arms only minimally and sporadically, consistent with the anxiogenic nature of the EPM paradigm.

To overcome this problem, we introduced a novel ELM on which rats were motivated to shuttle from one extremity to the other to receive rewards. The ELM had three different configurations: non-anxiogenic (CC configuration); anxiogenic (CO configuration); and a configuration with new texture and visual cues (CT). Configurations could be quickly switched within a session. At the behavioural level, we observed anxiety-related behaviour during ELM exploration comparable to the ones during EPM exploration (*Figure 2D and E*). However, the main advantage of the ELM was the possibility to record the neuronal activity of the same vH neurons while rapidly modifying ELM configurations and

motivating rats to spend more time in the anxiogenic location. After animals transited from a non-anxiogenic to an anxiogenic configuration, we observed a significant increment in peak firing activities predominately located in the open area. We attributed this recruitment or remapping of the neuronal activity to the anxiogenic location. The neuronal mechanisms during spatial remapping remain largely elusive. Global remapping is a phenomenon observed in the dH when animals move from one environment to a different one (*Leutgeb et al., 2005*). One could argue that the simple removal of the walls is changing the environment of the ELM and therefore animals could perceive the open arm as a completely novel environment, inducing global remapping. Nevertheless, remapping of neuronal activity in a new environment is expected to be random and independent of a previously explored environment (*Gauthier and Tank, 2018*; *Leutgeb et al., 2005*; *Schlesiger et al., 2018*), even when emotional contexts are introduced (*Moita et al., 2004*). However, the remapping of neuronal activity observed in vH neurons is not arbitrary as it is most prominent in the anxiogenic location, contrary to a uniformly distributed peak activity in case of a random remapping (*Figure 3B*). Nonetheless, the opening of walls per se, independent of the anxiogenic experience, could cause changes in vH neuronal activity. Previous work has shown that novel environments elicit the activation of vH neurons to a similar extent as an aversive stimulus (*Graham et al., 2021*). To show that the remapping of neuronal activity relies on anxiety, we changed the ELM from a CC to CT configuration during the same recording session. In the CT configuration, the previously open location changed to a novel one with a different floor texture and visual patterns in the inner part of the walls, but with protective sidewalls kept all along the track (*Figure 2B*). As anticipated, these changes equally induced a remapping of neuronal activity in the vH. Yet, in this case, the remapping was distributed over the length of the maze and the novel area did not show an increased number of peak firing activity (*Figure 3C*). Overrepresentation of behaviourally relevant locations or episodes is not novel in the hippocampal research field. It has been previously demonstrated that hippocampal place cells fire preferentially at reward locations during goal-directed tasks (*Dupret et al., 2010*; *Hok et al., 2007*; *Hollup et al., 2001*; *Jin and Lee, 2021*) and global remapping is observed in the intermediate hippocampus due to motivational changes (*Jin and Lee, 2021*). Several experiments have tackled the role of dH neurons in fear conditioning and other aversive tasks, resulting in similar activity patterns to the ones found in this research. Dorsal hippocampal place cells were shown to remap or partially remap in the presence of an aversive stimulus (*Kong et al., 2021*; *Moita et al., 2004*; *Wu et al., 2017*), or remap towards areas of aversive stimuli or threats (*Schuette et al., 2020*; *Wang et al., 2012*; *Wang et al., 2015*), by increasing their activation (*Wu et al., 2017*) or by synchrony with or reactivation of a subset of basal and BA neurons (*Girardeau et al., 2017*; *Kong et al., 2021*). Interestingly, the non-aversive stimulus, presented in the same context as the aversive ones, did not induce remapping in dorsal hippocampal neurons (*Moita et al., 2004*; *Wang et al., 2015*), a similar effect to our observations in vH neurons.

Furthermore, the vH is strongly associated with anxiety, and the anxiogenic location represents a highly salient environment for vH-dependent computations (*Bannerman et al., 2014*). This is supported by our finding that neurons of the vH overrepresent the anxiogenic area. Interestingly, in addition to the global remapping, we also found differences in the rate remapping between the CC-CO and the CC2-CT conformation changes. Rate remapping was significantly more robust in the novel context than in the anxiogenic one, in line with the literature suggesting that novelty induces rate remapping in hippocampal neurons (*Kaefer et al., 2019*; *Leutgeb et al., 2005*; *Rennó-Costa et al., 2010*). Regarding spatial properties of the hippocampal neurons, conflicting effects have been observed with aversive stimuli in the dH. In some cases, spatial information was not significantly different between non-aversive and aversive conditions, but the variance was significantly greater *Wu et al., 2017*; in other cases, spatial information was higher in non-aversive compared to aversive contexts *Wang et al., 2015*; while others report either an increase in place field size (*Moita et al., 2004*) or a decrease (*Schuette et al., 2020*). One paper reports various changes in the activity of vH neurons when mice are confronted with an aversive odour stimulus. They report a decrease in place field size, remapping of the activity and increase of the firing rate in the presence of the aversive odour (*Keinath et al., 2014*). Similarly, we also observe remapping of neuronal activity in aversive conditions and, in fact, a tendency, yet no significance, of the neurons to increase their firing rate in the CO configuration compared to the CC (*Figure 4—figure supplement 1C*). However, in our study, the activity of the neurons in vH increases its coverage, while in Keinath et al., it is reduced (smaller place fields). In the Keinath et al. study, the aversive stimulus was presented on a small area while in

our case the source of the anxiogenic stimulus cannot be pinpoint to a specific part of the open arm, which might have contributed to the different observations. In our experiments, the spatial properties of the neurons in the anxiogenic area of the CO configuration are reduced (lower spatial information and higher sparsity) compared to the CC configuration (*Figure 4*). Overall, all these elements further strengthen the role of vH in the emotional processing of information.

Another relevant observation relates to the directional firing of vH neurons. In the non-anxiogenic configuration of the ELM, the spatial activity of single vH neurons varied depending on the direction of exploration. Similar observations have been made in both the dH (*McNaughton et al., 1983*) and vH (*Royer et al., 2010*) for linear mazes, suggesting of a common principle underlying spatial information along the dorso-ventral axis of the hippocampus. With respect to the vH, Royer et al. hypothesised that the differential activity between inbound and outbound trajectories might be caused by the reward delivered at the end of the arm implying some reward-associated value coding. In contrast to Royer et al., we placed rewards at both extremities of the ELM. Although this does not invalidate the view of Royer et al., in our study, the direction-dependent firing was homogenised in the anxiogenic configuration of the ELM, with vH neurons exhibiting similar firing independently of the direction of exploration in the open arm (*Figure 5*). We attribute the homogenisation of the direction-dependent firing in single vH neurons to the relevance of the anxiogenic location for vH-dependent computations. One cannot rule out that the homogenisation is also a product of the noticeable change in the maze configuration, implying that the closed and open arm are easier to discern for the animal. Therefore, in both directions, the animal receives similar visual inputs and, therefore, activity homogenises. However, our control experiments, in which we introduced a new, but non-anxiogenic environment to the ELM (*Figure 5C*), also resulted in a clear differentiation of both parts of the ELM in both directions of travel. Still, we failed to observe a homogenisation of the activities.

Furthermore, the modulation of the neuronal activity in the anxiogenic location was not exclusively observed while the rats explored the open area. The CO configuration of the ELM contained protective walls in half of the maze, while the other half was entirely open. The neuronal population activity was a good predictor of the extent of the exploration of the open location, even when rats explored the closed arm before entering to the open location. Indeed, the neuronal activity in the closed arm, of neuronal pairs, was sufficient to infer whether rats would perform proximal or distal explorations of the open arm (*Figure 6*). Anxiety-related modulation of vH neuronal activity in both the closed and the open locations implies that not only a direct experience of anxiety enhances neuronal activity in the vH, but also its anticipation without a direct confrontation to an anxiety-inducing situation.

Overall, we provided evidence that the neuronal dynamics within the vH are subjected to the experience of anxiety. When an anxiogenic situation was encountered, vH neurons, first, over-represented this location (*Figure 3*). Second, spatial properties of the neurons, with their main activity in the anxiogenic area, were reduced (*Figure 4*). Third, vH neuronal activity was tuned to the anxiogenic environment, impairing previous direction-dependent neuronal activity manifesting in the absence of anxiety (*Figure 5*). Fourth, the neuronal activity of vH neurons reflected and predicted the exploration of an anxiogenic location (*Figure 6*). Collectively, these results expand our view of vH function by highlighting dynamic and predictive computations during anxiety.

## Materials and methods

### Experimental subjects

In total, nine long Evans rats from Charles River Laboratories (male, 250–600 g), were kept in 12 hr light cycles during behavioural experiments (performed during the light cycle). All experimental procedures were performed under an approved licence (66.009/0281-WF/V/3b/2015) of the Austrian Ministry of Science and the Medical University of Vienna.

### Surgery and microdrive implantation

Using isoflurane, animals were anaesthetised (induction 4%, maintenance 1–2%; oxygen flow 2 l/min) and fixed to a stereotaxic frame. The body temperature was controlled using a heating pad. Iodine solution was applied to disinfect the surgery site and eye cream was used to protect the eyes. Local anaesthetic (xylocain 2%) was used before the incision. Saline solution was injected subcutaneously every 2 hr, to avoid dehydration. Seven stainless steel screws were anchored into the skull to improve

the stability of the construct, and two of the screws were placed above the cerebellum as references for the electrophysiological recordings. Next, based on the rat brain atlas (*Paxinos and Watson, 2007*), a craniotomy was performed above the vH area (from bregma: –4.8 mm anterior, 4.5 mm lateral, right hemisphere). After removal of the dura mater, an array of 12 independently movable, gold-plated (100–500 kΩ) wire tetrodes (13 μm insulated tungsten wires, California Fine Wire, Grover Beach, CA) mounted in a custom-made microdrive (Miba Machine Shop, IST Austria) were implanted (dorso-ventral: –6.5 mm). Paraffin wax was then applied around the tetrode array, and the lower part of the microdrive was cemented (Refobacin Bone Cement) to the scalp. At the end, the surgery site was sutured, and systemic analgesia (metacam 2 mg/ml, 0.5 ml/kg) was given. Animals were allowed at least 7 days of recovery time.

## Histology

To confirm the position of the recording sites, rats were deeply anaesthetised with urethane and lesions were made at the tip of the tetrodes using a 30 μA unipolar current for 5–10 s (Stimulus Isolator, World Precision Instruments). Then, rats were perfused with saline followed by 20 min fixation with 4% paraformaldehyde, 15% (vol/vol) saturated picric acid and 0.05% glutaraldehyde in 0.1 M phosphate buffer. Serial coronal sections were cut at 70 or 100 μm with a vibratome (Leica). Sections containing a lesion were Nissl-stained. One rat, for which histological data could not be confirmed, was included based on: insertion coordinates, oscillatory LFP profile, and similarity of neuronal activity.

## Mazes description and behaviour

The EPM consisted of two closed (with protective sidewalls) and two open (without sidewalls) arms. The dimensions of the arms were 9×50 cm, the walls in the closed arms were 40 cm high, and the EPM was elevated 70 cm above the floor. Rats were placed on the EPM facing the open arm distal to the experimenter. Sessions lasted 5–8 min and were done at 200 lux of room light intensity (*Walf and Frye, 2007*). In total six animals (SC61, SC63, SC65, SC66, SC67, SC68) were used for the EPM. Some data of these animals are also used in another independent publication (*Forro et al., 2022*).

The ELM consisted of a linear track of 120 cm length and 8 cm in width. The maze was elevated by 105 cm above the ground. A reward was given at both endpoints (two 20 mg sugar pellets). Three possible configurations were presented during the EPM exploration: A CC configuration, consisting of four black panels acting as sidewalls which covered the entire length of the track and prevented the animal from experiencing the height; a CO configuration which consisted of two black panels acting as walls covering one half of the maze and leaving the other half completely open, resulting in an anxiogenic area; and a third configuration was called CT which consisted of four black panels acting as walls covering the entire length of the maze. The difference with the CC configuration was that in half of the maze (the half which was open in the CO configuration) coloured geometrical figures were added to the sidewalls as new visual patterns and the texture of the floor were changed. The ELM sessions were composed by the presentation of the CO and CT configurations, each preceded by a CC configuration. Depending on the motivation of animals to explore, CC and CT configurations lasted 5–15 min while CO configurations lasted between 5 and 20 min. When a CC configuration was preceding a CT configuration, we refer to the configuration as CC2.

Before animals were ready for a recording session, we habituated them to the CC configuration until they exhibited consistent shuttles between one end and the other (experimenter consideration). The animals did not experience CO and CT before recordings.

In total, 14 sessions were recorded from six rats in the ELM (SC65, SC67, SC68, SC70, HM15, HM16). CC or CC2 always preceded a CO or CT exploration, respectively. Configurations were distributed per animal in the following way: SC65, one session (CC, CO); SC67, two sessions (CC, CO); SC68, three sessions (CC, CO); SC70, three sessions (CC, CO); HM15: three sessions (CC, CO, CC2, CT); HM16: two sessions (CC, CO, CC2, CT). In three sessions, CT preceded CO, and in two others, CO preceded CT.

*Figure 2—figure supplement 1A, B* shows the boxplots for the number of trials, rewarded trials, and no-rewarded trials for each configuration. A trial was included as the trajectory from where an animal left a reward zone and crossed the middle. Only trials in which an animal went from one end to the other were rewarded. A repeated visit to the same reward zone from where the animal departed was not rewarded. However, if the animal crossed the middle before returning, it was counted as

two directional trials: one trial in one direction (used in the distal to proximal exploration analyses in *Figure 6*), and from the point of return, one trial in the other direction (used for the homogenisation analyses of *Figure 5*).

Tracking of the rats' movement was monitored by triangulating the signal from three LEDs (red, blue, green) placed on the implanted headstage and recorded at 50 frames per second by an overhead video camera (Sony).

## In vivo electrophysiology

Either an Axona headstage (HS-132A, 2 × 32 channels, Axona Ltd) or Intan headstage (2× RHD32-channel headstage) were used to pre-amplify the extracellular electric signals from the tetrodes. For the Axona headstages, output signals were amplified 1000× via a 64-channel amplifier and then digitised continuously with a sampling rate of 24 kHz at 16-bit resolution, using a 64-channel analogue-to-digital converter computer card (Axona Ltd). For the Intan headstages, signals were acquired with the RHD32 channel headstage and directly sent to the Intan 512ch/1024ch recording controller. Single-unit offline detection was performed by thresholding the digitally filtered signal (0.8–5 kHz) over five standard deviations (SD) from the root mean square in 0.2 ms sliding windows. For each single unit, 32 data points (1.33 ms) were sampled. A principal component analysis was implemented to extract the first three components of each spike waveform for each tetrode channel (*Csicsvari et al., 1998*).

Spike waveforms from individual neurons were detected using the KlustaKwik automatic clustering software (*Kadir et al., 2014*). Using the Klusters software (*Hazan et al., 2006*), single units were isolated manually by verifying the waveform shape, waveform amplitude across tetrode's channels, temporal autocorrelation (to assess the refractory period of a single unit) and cross-correlation (to assess a common refractory period across single units). The stability of single units was confirmed by examining spike features over time. The position of tetrodes was reset after each recording day. The following number of neurons were recorded for each animal: SC65, 2 neurons; SC67, 17 neurons; SC68, 31 neurons; SC70, 7 neurons; HM15, 44 neurons; and HM16, 32 neurons.

## Spike-shape classification of putative pyramidal cells and putative interneurons

The putative identity of the recorded hippocampal neurons has been assessed by examining the spike shape and the firing rate of the individual neurons (*Csicsvari et al., 1998*; *Forro et al., 2022*; *Sirota et al., 2008*). We averaged all the spikes of a clustered unit to obtain a spike shape of each neuron. We calculated the latency between the trough and the following peak (hyperpolarisation period) and the firing rate of each neuron (see *Figure 3—figure supplement 2B*). We sorted putative pyramidal cells with a trough-to-peak latency higher than 0.3 ms and firing rate lower than 20 Hz. Putative interneurons were defined as units with a trough-to-peak latency lower than 0.3 ms and firing rate higher or equal to 10 Hz. Several neurons did not fall into any of these two categories. They were included in the main analyses, but not when differentiating putative pyramidal cells and putative interneurons.

## SWR detection

SWR were detected in a semi-automatic matter from the local field potential of the tetrode with the highest ripple amplitude. We filtered the signal between 130 and 230 Hz and calculated the root mean square amplitude in 10 ms sliding windows. We calculated the mean and SD of all points in the trace. We set the threshold for ripple detection at 5 SD above the mean. The beginning and end of an SWR were marked in the 1 SD crossing (from the mean). A visual inspection excluded some detected SWR as artefacts; sometimes, boundaries were adjusted.

## Firing rate maps and trajectory linearisation

To compute firing rate maps found in *Figure 1*, bins of 10×10 cm$^2$ were created. For each bin, the total number of spikes was divided by the rat's occupancy (in seconds): the firing rate maps were smoothed by convolving them into two dimensions with a Gaussian low-pass filter. For the EPM, by using the geometry of the maze, the centre and the arms were defined. Trajectories were then found by demarcating the consecutive tracked positions going from the furthest point reached on the arm, to the furthest point reached on the next visited arm. To linearise this position, each two-dimensional

point (x, y) was projected to the directional vector describing the arm to which that point belongs. Each projection was made by using the following equation:

$$Projection = \frac{P_{x,y}.D_{arm}}{\|D_{arm}\|}$$

where $P_{x,y}$ is the position to be projected, $D_{arm}$ is the directional vector of the corresponding arm, and $\|D\|$ denotes the norm of the vector D. Each trajectory was composed of three parts: starting arm, centre, and ending arm. For the starting and ending arm, the activity was calculated over the space by dividing the total number of spikes on each linear bin (5 cm) over the occupancy (in seconds) on that particular bin. However, due to the different possible trajectories that the animal can follow in the centre, the activity there was divided into five fixed time bins. Then, the linear firing rate maps (activity in the starting, centre, and ending arm) were smoothed by convolving them with a 1D Gaussian function. Linear firing rate maps on the ELM were calculated by dividing the space into bins (2.5 cm each) and for each bin, the corresponding spikes of each neuron were summed and divided by the occupancy (in seconds).

## Spatial properties

Three different spatial properties were calculated (spatial information, sparsity, and coherence) based on *Alme et al., 2014*; *Muller and Kubie, 1989*; *Skaggs et al., 1996*; *Zhang et al., 2014*.

The spatial information measured how much information about the spatial location of the animal is contained within the activity of a cell (bits/spike):

$$\sum_{i=1}^{N} p_i \frac{\lambda_i}{\lambda} log_2 \frac{\lambda_i}{\lambda}$$

where:

i=bin
$p_i$=the probability of occupancy of the bin
$\lambda$ =overall firing rate
$\lambda_i$ = the mean firing rate of the bin

The sparsity (also called coverage) measured the compactness of the place field.

$$Sparsity = \frac{\left( \sum p_i \lambda_i \right)^2}{\sum p_i \lambda_i^2}$$

The variables of the equation are the same as for spatial information.

Coherence measures the extent to which the firing rate in a bin is predicted by the rates of the neighbouring bins. It can be related to how 'organised' the firing rate appears.

$$Z = 0.5 \, ln \left( \frac{1+R}{1-R} \right)$$

where R is the correlation between the raw spike activity and the smoothed spike activity on the ELM.

## Place field similarity

The PFS was calculated by using the Spearman correlation between the z-scored linear firing rate map of a neuron while the animal is moving in one direction and the z-scored linear firing rate map of the same neuron while the animal is moving in the other direction.

## General linear model and neuronal spike activity relation to speed

We modelled the spike activity of each neuron (for different ELM configurations) by using the instantaneous running speed of the animal at each moment: $Spk_t = \beta_0 + \beta_1 S_t$, where *Spk* is the number of spikes at a given time *t* and *S* is the instantaneous running speed at the same time *t*. Residuals of the model were used as the spike-associated activity of the neurons, controlled by the speed of the animal. Due to the correction, the firing rate of some bins might be negative when calculated. However, this does not affect the data analyses using z-score values.

## SVM classifier and neuronal selection

The exploration of the open arm on the ELM was divided into either proximal exploration or distal exploration using a threshold per session ($Threshold = \frac{Furthest\ spatial\ bin - Nearest\ spatial\ bin}{2}$). An SVM classifier with a linear kernel was used to determine if the extent of the exploration of a given trial (proximal or distal exploration) was possible. We calculated the firing rates of the neurons prior to the entrance to the open area in the CO configuration (from the moment the animal was heading towards the open arm, until the spatial bin before reaching the centre). Using this firing rate, in a 'one-leave out cross-validation' fashion, the identity of each trial was predicted by training the classifier with the neuronal activity of all the other trials. Using a wrapper-like method (*Kohavi and John, 1997*), we iteratively check the performance of the SVM for all possible combinations of neuronal pairs for each session. The neuronal pair that gave the highest performance was then assumed to be optimal.

Due to the fact that the distal and proximal trial distributions were not even, in order to determine if a given performance value was in fact better than random, we shuffle 1000 times the trial IDs (distal or proximal) and repeated the classification. Only performance values above the 95% percentile of the shuffle distribution were considered as successful.

The z-score activity in the examples of *Figure 6B* were 'scaled'. This scaling is a parameter of the linear SVM that divides all the elements of the predictor matrix. Values of the predictors were then modified by this scale.

Two recording sessions were excluded due to either a low number of co-recorded cells (n<3) or insufficient exploration of the maze.

## Density of peak firing rates and bootstrapping

The density of the firing rate peaks on each of the six possible paths of the EPM exploration was calculated by summing the number of peaks in the same bin divided by the number of neurons with any activity in that bin. For example, if bin 10 has 20 peaks and 100 neurons are active in that bin, the density is 0.2; on the contrary, if bin 20 has 5 peaks but only 10 neurons are active in that bin, the density would be 0.5.

To determine the significance of the density distribution, we used a bootstrapping/shuffle method. Bootstrapping for *Figure 1E* was achieved by randomly reassigning the peak activity for each neuronal spatial map (average activity in the session) to any visited bin. A new density plot with the randomly assigned data was calculated. This procedure was performed 1000 times, and we applied a one-tailed test with a 95% confidence interval of the shuffled distribution. The grey area in *Figure 1— figure supplement 1B* shows the area between the median and the 95% percentile of the shuffled distribution.

## Statistical analyses

All calculations were made in MATLAB (Mathworks, version R2015b and R2019b) and statistical analyses were performed with MATLAB and Microsoft Excel. All the statistical tests used in this manuscript were non-parametric unless stated otherwise. Raw data was visualised and visually evaluated with Neuroscope (http://neurosuite.sourceforge.net/information.html).

One-way ANOVA with Tukey-Kramer for multiple comparison was used for the time spent on each of the EPM and ELM areas (*Figure 1A*, *Figure 2E*). Two-sided Wilcoxon signed-rank test was used for: time spent in changing and non-changing arms (*Figure 2D*); spatial properties (*Figure 4A and B*); and firing rates in different ELM configuration (*Figure 4—figure supplement 1C*). Chi-squared test was used for: proportions of peak firing activities (*Figure 3B, D, E and F* and *Figure 3—figure supplement 3A, B, C*). Because the chi-squared test depends on what is consider the expected proportion, it was done in two different ways. In the figures, the significant values are presented assuming a 50%–50% chance. We also tested the proportions of the peak locations in the CO and CT configurations, by assuming the expected distributions as the distribution obtained in the CC configuration. Significances in the *Figure 3* were the same in both assumptions. Changes in the significance levels are reported in the figure legend of the *Figure 3—figure supplement 3*. Two-sided Wilcoxon rank-sum test was used for *Figure 4C*, *Figure 4—figure supplement 1B, D*; comparing the PFS index distributions (*Figure 5B, C, D and E* and *Figure 5—figure supplement 1A, B, C*); and to compare the animal speeds in *Figure 6—figure supplement 1B*.

Analyses of CC and CO have an n=133, while analyses of CC2 and CT have an n=75. Both numbers correspond to the recorded neurons in the vH. Time spent measurements were done per session: EPM sessions (n=16), ELM sessions (n=14), control sessions of CC2 and CT (n=5).

To corroborate that the results were not driven by the data of a particular rat, we checked, separately per animal, the distributions of spatial information, sparsity, coherence, directional correlation of the neuronal activity, and the distributions of neuronal peak firing locations for the CC and the CO configurations. No significant differences arose between different animals, except for the correlation values for the CC configuration in rat SC68, in comparison with other two animals. We confirmed that even after removing the data of SC68 from the data set, all the reported results of the research were still maintained.

## Lead contact

Further information and requests may be directed to the Lead Contact, Thomas Klausberger (thomas. klausberger@meduniwien.ac.at).

## Acknowledgements

We thank R Hauer for her excellent technical support; M Nigritinou for histological help; C Espinoza for comments on the figures; B Lasztoczi and P Anderson for comments on the analyses. This work was supported by grant P 29588 of the Austrian Science Fund (TK), ERC starting grant 716761 (SC), and a Swiss National Science Foundation professorship grant (170654) (SC).

## Additional information

### Funding

| Funder | Grant reference number | Author |
|---|---|---|
| Austrian Science Fund | P 29588 | Thomas Klausberger |
| European Research Council | 716761 | Stéphane Ciocchi |
| Swiss National Science Foundation | 170654 | Stéphane Ciocchi |

The funders had no role in study design, data collection and interpretation, or the decision to submit the work for publication.

### Author contributions

Hugo Malagon-Vina, Conceptualization, Data curation, Software, Formal analysis, Validation, Investigation, Visualization, Methodology, Writing – original draft, Writing – review and editing; Stéphane Ciocchi, Conceptualization, Resources, Data curation, Software, Formal analysis, Supervision, Funding acquisition, Validation, Investigation, Methodology, Writing – original draft, Project administration, Writing – review and editing; Thomas Klausberger, Conceptualization, Resources, Data curation, Formal analysis, Supervision, Funding acquisition, Validation, Investigation, Methodology, Writing – original draft, Project administration, Writing – review and editing

### Author ORCIDs

Hugo Malagon-Vina ⓘ http://orcid.org/0000-0001-5029-8464
Thomas Klausberger ⓘ http://orcid.org/0000-0001-7269-3158

### Ethics

All experimental procedures were performed under an approved licence (66.009/0281-WF/V/3b/2015) of the Austrian Ministry of Science and the Medical University of Vienna.

### Decision letter and Author response

Decision letter https://doi.org/10.7554/eLife.83012.sa1
Author response https://doi.org/10.7554/eLife.83012.sa2

## Additional files

### Supplementary files
• Transparent reporting form

### Data availability
Data is available in Github (https://github.com/Klausberger-Laboratory/Malagon-Vina-Anxiety2022; copy archived at *Malagon-Vina, 2023*).

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
