## [Editor Report]

This paper is expected to be of interest to systems neuroscientists in the fields of emotion, hippocampal function, and anxiety-related behavior. The authors performed recordings in the ventral hippocampus and show that (1) place fields become concentrated near the open areas of a maze, (2) direction-dependent coding decreases in these open areas, and (3) ventral hippocampal population activity in the closed area can be used to predict how mice explore the open area in the immediate future. These valuable findings provide convincing support for the potential role of the ventral hippocampus in the exploration of anxiety-provoking environments.

---

## [Decision Letter]

**Decision letter after peer review:**

Thank you for submitting your article "Firing patterns of ventral hippocampal neurons predict the exploration of anxiogenic locations" for consideration by *eLife*. Your article has been reviewed by 3 peer reviewers, and the evaluation has been overseen by Laura Colgin as the Senior Editor. The following individual involved in review of your submission has agreed to reveal their identity: Inah Lee (Reviewer #1).

Essential revisions:

1) Necessary neural analyses are missing from the paper. For example, basic analyses of single cell properties are lacking.

2) Previous dorsal hippocampal studies related to processing anxiety and fear need to be discussed more deeply to put the new findings within the proper context.

3) Behavioral data are not described and analyzed sufficiently. As it stands, it is unclear whether neural effects are due to differences in behavior.

4) Putative principal cells and interneurons should be analyzed separately, and it should be clear how cell types were subdivided.

5) Some details about methods are missing. For example, how were spikes during sharp wave-ripples removed?

6) Histological verification should be shown.

Please see individual reviews below for details and specific recommendations.

*Reviewer #1 (Recommendations for the authors):*

The primary finding of this study is that it was possible to predict the extent of exploration of the anxiogenic area by using the neural activities of the ventral hippocampus (vHP) before the rat enters such an area (i.e., the open arm) (Figure 5). However, the authors did not provide detailed neural and behavioral data to support their arguments. First of all, to verify the electrophysiological data, they need to report the basic firing properties of single-cell activities and the representative histological photomicrographs showing the electrode tips to verify whether the electrodes were indeed targeted the pyramidal cell layers in the vHP (e.g., the overall distribution of the spike width and mean firing rate during CC, CO, CT condition, raw spiking samples to show cell's recording stability, brain sections with thionin staining, etc.). In addition, the authors need to clarify how pyramidal neurons and interneurons were distinguished from each other (e.g., using the mean firing rate and spike width?), and how they removed sharp-wave ripple-associated spikes (e.g., using a speed filter?). Second, before providing results of principal component analysis (PCA) and support vector machine (SVM), they need to verify that the mean firing rate before entering the open arm was positively or negatively correlated with the exploration types (i.e., proximal and distal exploration). Such correlation analysis may make the results of PCA and SVM more substantial.

Next, the current version of the manuscript lacks detailed behavioral data (e.g., velocity and position differences between proximal and distal exploration trials before entering the open arm), which may raise the alternative hypothesis that the difference in the neural activities between proximal and distal exploration could result from behavioral differences, not the prediction signals in the vHP. For example, in the proximal exploration trials, rats could be more hesitant and stayed longer near the boundary before entering the open arm than in the distal exploration trials. In contrast, in the distal exploration trials, rats might run toward the rewards in the open arm without hesitation. In that case, it is possible that the difference in neural activities between the exploration types could mainly reflect the difference in animal speed or how long they stayed near the boundary, not necessarily reflecting the animal's intention to explore the anxiogenic area. To address these concerns, the authors need to report detailed behavioral and neural data with respect to the issue. For example, they may need to provide a peri-event time histogram of the animal's velocity and position and spikes in relation to the time of entering the open arm. Alternatively, as they used spike-associated activity controlled by speed via the general linear model (GLM) in Figure 3F-3G and 4D-4E, they may apply this model to this analysis in Figure 5 to control the speed factor, etc, to name a few.

Also, it is unclear which neural firing data were used for the PCA and SVM analyses. The authors said "firing rate of each neuron prior to the open arm ~ (line 478)", but it is confusing which moment they indicated by 'prior.' They need to quantitatively define 'prior' (e.g., activities within seconds before entering the open arm or activities within 10cm from the boundary between the open and closed arm). If position or velocity was different depending on the exploration type, it could be that different place cells might be recruited during the "firing rate prior to the open arm," which might result in distinct neural activities between proximal and distal exploration trials. In addition, they need to report how PCs in proximal and distal exploration were quantitatively separated between two exploration types in the entire session, not only for two sessions (Figure 5A).

In SVM analysis, the authors did not show raw data to demonstrate that the neural activities in proximal and distal exploration trials were properly distinguished through the hyperplane of SVM (Figure 5B). Thus, it is impossible to assess the validity of SVM analysis. The authors need to provide the graph whose x and y axes are associated with the neuronal activities and demonstrate that the hyperplane properly distinguished proximal and distal exploration trials. Moreover, they iteratively computed the performance of the SVM for all possible combinations of neurons and chose the combination that gave the highest performance for further analysis (Figure 5B – 5C). This may result in an overestimation of the results. For example, even if most combinations had low performance, it would be considered a high-performance session if there was only one combination of higher performance. Thus, the authors need to provide the results of all combinations. Also, it seems more appropriate to infer performance using the ensemble data rather than based on a combination of two cells.

*Reviewer #2 (Recommendations for the authors):*

Related suggestions/comments to the Public Review Weakness points:

Related suggestion to W1.

It would be helpful if the authors could make a clear comparison between the dorsal and ventral hippocampus with regard to anxiety responses and highlight what aspects are unique to the vH anxiety driven cells.

Related suggestion to W2.

It would be helpful for readers to know which subpopulation of cells in vH function as the basis for dynamical remapping of anxiety information.

Additionally, it would be helpful to have a table describing how many neurons were recorded from each animal and from what subregions. In Figure 1B, many red dots are missing the cell layer of the vH, especially in the third panel and readers may wonder where the neural data was actually recorded from.

Other specific points:

In line 251

No PCA descriptions in the Methods section.

In lines 399-403

In total, how many sessions did each animal perform per day?

Error bars are missing in Figure 3B, 3F.

In Figure 4A

The representative 'homogenized' cells # 41 and 110 are confusing as the firing rates in closed area (spatial bins up to 10) seem also 'homogenized'. One would expect to see the cells that are only 'homogenized' in the open area (spatial bins between 10-20), no?

In Figure 5A

What does "n=11" and "n=3" indicate here, animals, trials, or neurons?

In lines 511 and 512

While there are descriptions "Figure S1A, B" and "Figure S1C, D", there are no supplemental figures provided with the manuscript.

*Reviewer #3 (Recommendations for the authors):*

1. The authors should include in their discussion the results from Wang et al., 2012 and Wang et al., 2015 (PMID: 26085635 and PMID: 23136419). The authors also should discuss Kong et al., 2021 (PMID: 34533133) and Schuette et al., 2021 (PMID: 32958567). Kong found more remapping near the threat, while Schuette found a concentration of place field centers near the threat and a decrease in place field size near the threat. These results were different because they done with dorsal hippocampal cells and different types of threats, but these data must be discussed to put the new findings within the proper context.

2. The authors show that after the cc to co transition there are more cells with peak firing rates in the open area. Prior to the transition, is there any difference between the cells that moved their peak firing rate location to the open area compared to the ones that did not move? Are there differences in prior firing rate, field size, spatial information during cc that predict the cell's remapping in co?

3. Comparing the cells that lost directionality and the ones that did not, was there any difference in these cells after or during the transition? Are there differences in prior firing rate, field size, spatial information during cc that predict the cell's loss of directionality in co?

4. In Figure 3d why do the authors use 'peak location' instead of place field center? What would the data look like with place field center plotted instead?

5. Did the data change in any way across the multiple sessions of recordings? Did the results about coverage, directionality and concentration of peak locations in the open area change across the multiple sessions?

6. Please show the main effects separated by animal sex.

7. Provide examples of real histology photos instead of diagrams.

8. Add experimental details showing for each animal how many sessions in ELM were obtained, how many trials and across how many days.

9. I am having trouble understanding what is plotted in Figure 5c. The legend says "Box plots show median, 25th and 75th percentile". But it is the median of what? What is plotted? Is it possible to plot these data in some other way?

[Editors’ note: further revisions were suggested prior to acceptance, as described below.]

Thank you for resubmitting your work entitled "Firing patterns of ventral hippocampal neurons predict the exploration of anxiogenic locations" for further consideration by *eLife*. Your revised article has been evaluated by Laura Colgin (Senior Editor) and Reviewers.

The manuscript has been improved but there are some remaining issues that need to be addressed, as outlined below:

Please see the remaining issues noted by Reviewers #1 and 3 below.

Summary:

This paper is expected to be of interest to systems neuroscientists in the fields of emotion, hippocampal function, and anxiety-related behavior. The authors performed recordings in the ventral hippocampus and show that (1) place fields become concentrated near the open areas of a maze, (2) direction-dependent coding decreases in these open areas, and (3) ventral hippocampal population activity in the closed area can be used to predict how mice explore the open area in the immediate future. These valuable findings provide convincing support for the potential role of the ventral hippocampus in the exploration of anxiety-provoking environments.

*Reviewer #1 (Recommendations for the authors):*

The authors have addressed most of my concerns. However, there are still some concerns that may need to be addressed (see below).

– (Figure 3A) The authors nicely illustrated single-cell examples in Figure 3 (and figure supplement 1A and 1B) to show broader place fields in the ventral hippocampus. However, when examining the individual cases, I am concerned that the z-transformed population rate maps in Figure 3A may give the reader the wrong impression that most cells in the ventral hippocampus have focal place fields. Furthermore, inhibitory interneurons in the hippocampus also have their preferred firing locations where their firing rates are higher than others (Ego-Stengel and Wilson, 2007, Hippocampus). As the current study didn't differentially analyze putative pyramidal cells and interneurons, it might be difficult to distinguish between activities in pyramidal neurons and interneurons when using the z-transformed rate maps. Therefore, it seems inappropriate to show the population activities using a z-transformed population rate map.

– (Figure 6 —figure supplement 1B) The authors used the averaged speed in the arena spanning 15 cm before crossing the open area to argue that there was no behavioral difference between proximal and distal exploration. However, 15 cm is an arbitrarily determined value, and I wonder if it can serve as a representative measurement to compare animal behavior between proximal and distal trials. This is mainly because they calculated firing rates for SVM analysis from the moment heading towards the open area, not the 15cm before crossing the boundary between closed and open areas. Thus, it might be much more appropriate to compare the speed from the moment heading toward the open area to the end of the closed area.

– Additionally, the authors should include statistical testing in Figure 6 —figure supplement 1B. Based on my observations, the speed of future proximal exploration seems significantly lower than that of future distal exploration in session 5. Along with the results of each session, it might be necessary to pool the data from all sessions and perform statistical testing in order to argue that there was no speed difference between proximal and distal exploration.

– (Discussion) Keinath and colleagues (Hippocampus, 2014) argued that cells in the mouse ventral hippocampus showed more spatially selective firing patterns when aversive odor (i.e., predator's urine) was introduced in the open arena. This result contradicts the current manuscript because adding an anxiety factor seemed to decrease spatial firing characteristics in the current study as opposed to the results of Keinath et al. (2014). If the authors explain the potential factors of why there was a contradiction between Keinath and colleagues and the current study, it will be helpful to understand the importance of the ventral hippocampus in emotional information processing. Additionally, if the authors explain why the firing fields became larger but not smaller in anxiogenic space in processing emotional processing, it can give readers a clue about how the ventral hippocampus is involved in processing emotional information.

*Reviewer #2 (Recommendations for the authors):*

After carefully evaluating the revised manuscript and the authors' response to my previous comments, I am pleased to report that I am satisfied with the changes made by the authors. The authors have addressed all the concerns and/or issues raised in my previous review and have made significant improvements to the manuscript.

The revised manuscript, with the additional supplemental materials, now presents a clear and concise argument, and the additional data analysis, especially on the single unit analyses and presentation has been significantly strengthened.

The author has also provided additional information (incl. Data Availability) and clarification where necessary, which has improved the quality of the manuscript.

Overall, I recommend that the revised manuscript is now suitable for publication in eLife.

*Reviewer #3 (Recommendations for the authors):*If the authors used only male mice this fact must be stated in the abstract.

---

## [Author Response]

Essential revisions:1) Necessary neural analyses are missing from the paper. For example, basic analyses of single cell properties are lacking.

We recognise that the manuscript should have presented some fundamental analyses of single-cell properties. Accordingly, we have added new data regarding this concern:

– Examples of the averaged firing rates for different cells in different configurations have been added in Figure 3 —figure supplement 1

– Spike-shape stability over time is shown for all the neurons in Figure 3 —figure supplement 2

– Spatial properties (spatial information, sparsity and coherence) of the neurons are shown in the new Figure 4 and Figure 4 —figure supplement 1

– Firing rates of the neurons and rate remapping properties between different ELM configurations are presented in Figure 4 —figure supplement 1

2) Previous dorsal hippocampal studies related to processing anxiety and fear need to be discussed more deeply to put the new findings within the proper context.

We thank the referees for the recommendation. We followed this advice and the suggestions have improved the discussion (page 12 – 13, lines 370 – 398). We have also added the suggested citations.

3) Behavioral data are not described and analyzed sufficiently. As it stands, it is unclear whether neural effects are due to differences in behavior.

We thank the reviewer for the very useful comment, especially regarding the results for the prediction of further exploration. We agree that speed differences could have explained the neuronal activity differences. We have added a series of analyses and clarifications:

– We have improved the description of the behavioural assays in the methods, paying particular attention to the request of the individual reviewers.

– We performed control analyses for several experiments and corrected for speed-related neuronal activity. In particular, we are now also showing the results of the SVM classifiers, predicting the proximal or distal future exploration of the subject, for the speed-corrected firing (Figure 6 —figure supplement 2B).

– In addition, we show that the actual speed of the animal, prior to the entrance of the open area in the CO configuration, is not significantly different between future proximal or distal explorations (Figure 6 —figure supplement 1A).

– Another behavioural difference that might affect the neuronal activity of hippocampal neurons is their association with periods of sharp-wave ripples (SWR). We have now detected SWR, removed the spikes associated with those periods and re-analysed the data showing that our conclusions still hold for the SWR-corrected activity (Figure 3 —figure supplement 3C; Figure 5 —figure supplement 1C and Figure 6 —figure supplement 2B).

4) Putative principal cells and interneurons should be analyzed separately, and it should be clear how cell types were subdivided.

We thank the reviewers for this suggestion. We followed the advice and separated putative pyramidal neurons from putative interneurons using their neuronal firing rate and spike width (Figure 3 —figure supplement 2B). Using this separation, we performed all of the analyses also separating put. pyramidal cells and put. interneurons. Interestingly, both group of cells show very similar activity pattern changes, with few specific differences highlighted now in the text (Figure 3 —figure supplement 3A, B); (Figure 5 —figure supplement 1A, B; Figure 6 —figure supplement 2B).

5) Some details about methods are missing. For example, how were spikes during sharp wave-ripples removed?

We acknowledge the question of the reviewers. We have now improved the details of this analysis in the methods and Results section.

6) Histological verification should be shown.

We agree with the reviewers and have added images of several Nissl-stained brain sections with the tetrodes targeting the ventral CA1 area (Figure 1 —figure supplement 1).

Reviewer #1 (Recommendations for the authors):The primary finding of this study is that it was possible to predict the extent of exploration of the anxiogenic area by using the neural activities of the ventral hippocampus (vHP) before the rat enters such an area (i.e., the open arm) (Figure 5). However, the authors did not provide detailed neural and behavioral data to support their arguments. First of all, to verify the electrophysiological data, they need to report the basic firing properties of single-cell activities and the representative histological photomicrographs showing the electrode tips to verify whether the electrodes were indeed targeted the pyramidal cell layers in the vHP (e.g., the overall distribution of the spike width and mean firing rate during CC, CO, CT condition, raw spiking samples to show cell's recording stability, brain sections with thionin staining, etc.).

We thank the reviewer for the suggestions. We have added pictures of Nissl-stained histological brain sections with tetrode tracks in ventral hippocampus (Figure 1 —figure supplement 1). In addition, we have performed an analysis to corroborate the stability of the recordings across the sessions (Figure 3 —figure supplement 2A). In the analyses, we calculated the averaged spike shape for single neurons during the first third of the recording day and correlated that spike with the average shape of the last third. 100% of the spike-shapes had a minimum correlation value of 0.98, implying that the spikes were identical at the beginning and the end of the recording. Also, we have now provided examples of the average activity for different neurons in different ELM configurations (Figure 3 —figure supplement 1), firing rates during different configurations (Figure 4 —figure supplement 1C) and the raster plots of the neuronal data used for the SVM classification (Figure 6 —figure supplement 1A).

In addition, the authors need to clarify how pyramidal neurons and interneurons were distinguished from each other (e.g., using the mean firing rate and spike width?),

We have added the differentiation between putative pyramidal cells and putative interneurons and re-analysed the data separating these two populations. We have used both the firing rate and the spike width (Figure 3 —figure supplement 2B; Figure 3 —figure supplement 3A, B; Figure 5 —figure supplement 1A, B; Figure 6 —figure supplement 2B; and see methods).

and how they removed sharp-wave ripple-associated spikes (e.g., using a speed filter?).

We have added the SWR detection to the methods, and we have re- analysed the neuronal activity after removing spikes associated with SWR periods (Figure 3 —figure supplement 3A; Figure 5 —figure supplement 1C; Figure 6 —figure supplement 2B). This did not change our conclusions.

Second, before providing results of principal component analysis (PCA) and support vector machine (SVM), they need to verify that the mean firing rate before entering the open arm was positively or negatively correlated with the exploration types (i.e., proximal and distal exploration). Such correlation analysis may make the results of PCA and SVM more substantial.

We followed the advice of the referee. We were previously using the PCA just as a visualisation tool to imply that the population was separating the two explorations based on the neuronal firing rate prior to entering the open arm. We have changed that, and we have decided to show the prediction of the exploration based on the activity of neuronal pairs. We plot examples of the neuronal firing rates, one with the neuronal firing rate per trial for the proximal and distal explorations (Figure 6B) and another as raster plots of those neurons' activity before entering the open arm (Figure 6 —figure supplement 1A). In some cases, for example, in the example of session 2 (Figure 6B), you see that Neuron 8 (N8) activity is correlated with the explorations, but neuron 3 (N3) is not so much. Also, in the neuronal activity in example session 9, none of the two neurons entirely correlate with the exploration. Nevertheless, the advantage of the classifier is that it can still separate both explorations (distal and proximal) based on the combination of both neuronal firing rates.

Next, the current version of the manuscript lacks detailed behavioral data (e.g., velocity and position differences between proximal and distal exploration trials before entering the open arm), which may raise the alternative hypothesis that the difference in the neural activities between proximal and distal exploration could result from behavioral differences, not the prediction signals in the vHP. For example, in the proximal exploration trials, rats could be more hesitant and stayed longer near the boundary before entering the open arm than in the distal exploration trials. In contrast, in the distal exploration trials, rats might run toward the rewards in the open arm without hesitation. In that case, it is possible that the difference in neural activities between the exploration types could mainly reflect the difference in animal speed or how long they stayed near the boundary, not necessarily reflecting the animal's intention to explore the anxiogenic area. To address these concerns, the authors need to report detailed behavioral and neural data with respect to the issue. For example, they may need to provide a peri-event time histogram of the animal's velocity and position and spikes in relation to the time of entering the open arm. Alternatively, as they used spike-associated activity controlled by speed via the general linear model (GLM) in Figure 3F-3G and 4D-4E, they may apply this model to this analysis in Figure 5 to control the speed factor, etc, to name a few.

We thank the reviewer for the clever observation. We did not account for it previously, and therefore we have performed two different analyses in the direction of the suggestion:

– We show that the speed of the animal prior to the entrance to the open arm (15 cm) is not significantly different between proximal and distal explorations. (Figure 6 —figure supplement 1B).

– We were also able to predict the type of exploration (Distal or Proximal) in several sessions, using the spike-associated activity corrected by speed as suggested by the referee (Figure 6 —figure supplement 2B).

Also, it is unclear which neural firing data were used for the PCA and SVM analyses. The authors said "firing rate of each neuron prior to the open arm ~ (line 478)", but it is confusing which moment they indicated by 'prior.' They need to quantitatively define 'prior' (e.g., activities within seconds before entering the open arm or activities within 10cm from the boundary between the open and closed arm).

We apologise for the lack of clarity regarding the period during which firing rates were calculated for the SVM analyses. We have now clarified in the methods section that we took the data from the moment the animal headed towards the open area until the last bin covering the closed area. This time varies depending on the decision of the animal to start approaching the open area.

If position or velocity was different depending on the exploration type, it could be that different place cells might be recruited during the "firing rate prior to the open arm," which might result in distinct neural activities between proximal and distal exploration trials.

We thank the reviewer for raising this possibility. We are now showing the results of the SVM classifiers, predicting the proximal or distal future exploration of the animal and controlling for speed-related associated firing (Figure 6 —figure supplement 2B). In addition, we show that the actual speed of the animal, prior to the entrance of the open area in the CO configuration, is not significantly different between future proximal or distal explorations (Figure 6 —figure supplement 1A).

In addition, they need to report how PCs in proximal and distal exploration were quantitatively separated between two exploration types in the entire session, not only for two sessions (Figure 5A).

We apologise for the confusion caused by not correctly labelling figure 5 (now figure 6). We are showing two examples (session 2 and session 9) in which the activity of a pair of neurons separates proximal and distal exploration (Figure 6B. We are no longer using PCA). However, Figure 6C shows three other sessions where the separation is possible, and the performance of the SVM is high. In addition, Figure 6D shows all the sessions (12 in total) and how the SVM can, better than shuffled data, discern between proximal and distal exploration in 9 of those sessions (green dots). We have improved the explanations of the figures and corrected the y-axis of Figure 6D.

In SVM analysis, the authors did not show raw data to demonstrate that the neural activities in proximal and distal exploration trials were properly distinguished through the hyperplane of SVM (Figure 5B). Thus, it is impossible to assess the validity of SVM analysis. The authors need to provide the graph whose x and y axes are associated with the neuronal activities and demonstrate that the hyperplane properly distinguished proximal and distal exploration trials. Moreover, they iteratively computed the performance of the SVM for all possible combinations of neurons and chose the combination that gave the highest performance for further analysis (Figure 5B – 5C). This may result in an overestimation of the results. For example, even if most combinations had low performance, it would be considered a high-performance session if there was only one combination of higher performance. Thus, the authors need to provide the results of all combinations.

We have changed the presentation according to the suggestions and added new plots to address this:

– Figure 6B shows the x and y-axis with the z-score activity of the neuron during the session for the different trials.

– Figure 6 —figure supplement 1A shows raster plots of the neuronal activity.

– Figure 6 —figure supplement 2A shows in red boxplots the performance (differential to the optimal) of the rest of the population pairs in comparison with the selected pair (green line) and the shuffle data (black boxplots).

Also, it seems more appropriate to infer performance using the ensemble data rather than based on a combination of two cells.

We create models with pairs of neurons instead of the entire population because of the possible overfitting that can be created when the number of observations is less than the number of variables used (e.g. 15 trials in total but 18 neurons). The heuristics in the number of necessary observations per variable are relatively subjective. However, a used rule of thumb is approximately 10 – 20 observations per variable (Valliappa Lakshmanan, Sara Robinson, and Michael Munn. *Machine learning design patterns*. O’Reilly Media, 2020). In our case, that will imply that a maximum of 2 neurons is closer to the best model to avoid overfitting. Then we used a wrapper-like method to determine which neuronal pair best predicted the exploration ID (proximal, distal). To avoid overfitting regarding this approach, we do not use the entire observations for the model and then determine a performance. In one-leave-out cross-validation, we predict each trial with the model trained in the other ones. The prediction of each trial is then used to calculate the final performance score per neuronal pair.

Reviewer #2 (Recommendations for the authors):Related suggestions/comments to the Public Review Weakness points:Related suggestion to W1.It would be helpful if the authors could make a clear comparison between the dorsal and ventral hippocampus with regard to anxiety responses and highlight what aspects are unique to the vH anxiety driven cells.

We thank the referee for the positive evaluation of our manuscript and we agree with the reviewer's suggestion and have added dorsal hippocampus research to the discussion, in which aversive stimuli have been used.

Related suggestion to W2.It would be helpful for readers to know which subpopulation of cells in vH function as the basis for dynamical remapping of anxiety information.Additionally, it would be helpful to have a table describing how many neurons were recorded from each animal and from what subregions. In Figure 1B, many red dots are missing the cell layer of the vH, especially in the third panel and readers may wonder where the neural data was actually recorded from.

We thank the reviewer for the suggestion***.*** The numbers of neurons per animal have been added in the methods section. In addition, we have separated the populations into putative pyramidal cells and putative interneurons (Figure 3 —figure supplement 2B) and re analysed the data to have a better understanding of how the sub-populations relate to the anxiety information (Figure 3 —figure supplement 3A, B; Figure 5 —figure supplement 1A, B; Figure 6 —figure supplement 2B). Interestingly, only minor differences were found in the differentiation. Those are discussed in the Discussion section. Regarding the anatomical subregion from where the neural data was recorded, because of the overlap of pyramidal cells between CA1 and subiculum, we refrain from giving an exact number of location-specific recordings.

Other specific points:In line 251No PCA descriptions in the Methods section.

Based on the suggestions and insight of the entire revision, we have decided not to use the PCA in the revision as it did not add information to the findings.

In lines 399-403In total, how many sessions did each animal perform per day?

One session per recording day. We have added the number of sessions per animal in the methods sections.

Error bars are missing in Figure 3B, 3F.

We apologise for the confusion. We have now included the clarification that we have absolute proportions, in total and not per session. Therefore there are no error bars in those plots.

In Figure 4AThe representative 'homogenized' cells # 41 and 110 are confusing as the firing rates in closed area (spatial bins up to 10) seem also 'homogenized'. One would expect to see the cells that are only 'homogenized' in the open area (spatial bins between 10-20), no?

We acknowledge the confusion created by the lack of clarification in our results. We consider homogenisation as the activity being similar over the entire arm and not compartmentalised in a specific area (for example, the open area). However, the fact that there is homogenisation and an increased number of peak firing activities located in the open arm has led us to conclude that this is more specific for the open area. We do not dare to make that claim due to the entangled interactions of the observations in that respect.

In Figure 5AWhat does "n=11" and "n=3" indicate here, animals, trials, or neurons?

We meant neurons, but we have now changed that figure.

In lines 511 and 512While there are descriptions "Figure S1A, B" and "Figure S1C, D", there are no supplemental figures provided with the manuscript.

We apologise for the lack of attention with the submitted file. We have corrected it.

Reviewer #3 (Recommendations for the authors):1. The authors should include in their discussion the results from Wang et al., 2012 and Wang et al., 2015 (PMID: 26085635 and PMID: 23136419). The authors also should discuss Kong et al., 2021 (PMID: 34533133) and Schuette et al., 2021 (PMID: 32958567). Kong found more remapping near the threat, while Schuette found a concentration of place field centers near the threat and a decrease in place field size near the threat. These results were different because they done with dorsal hippocampal cells and different types of threats, but these data must be discussed to put the new findings within the proper context.

We thank the reviewer for the positive evaluation and highlighting the mentioned papers, which improved the introduction and discussion and have added these citations to the manuscript.

2. The authors show that after the cc to co transition there are more cells with peak firing rates in the open area. Prior to the transition, is there any difference between the cells that moved their peak firing rate location to the open area compared to the ones that did not move? Are there differences in prior firing rate, field size, spatial information during cc that predict the cell's remapping in co?

We thank the reviewer for the idea of checking the spatial properties prior to the CO configuration for those cells that are remapping. We have added the results in Figure 4 —figure supplement 1A. Interestingly, it seems that the spatial information and the sparsity of the neurons that will remap to the open areas are significantly different from those that will not.

3. Comparing the cells that lost directionality and the ones that did not, was there any difference in these cells after or during the transition? Are there differences in prior firing rate, field size, spatial information during cc that predict the cell's loss of directionality in co?

The comparison is tricky to make because of the problem of defining neurons that lost or gained directionality. Being not a clear threshold to account for that, we are not separating direction-related neurons from not-direction-dependent ones. What we are showing is that the activity is significantly more similar for both directions when the CO configuration is explored in contrast to the CC.

4. In Figure 3d why do the authors use 'peak location' instead of place field center? What would the data look like with place field center plotted instead?

We apologise for the confusion. We are using the “place field” peak, which is the peak of the already averaged and Gaussian windowed activity. In most cases, these two are the same. There might be some differences when using the “place field centre” in neurons whose activity is not place related per se. Then, in that case, the “centre of mass” of the activity can be misleading to the actual peak activity. We then decided to use and call it peak location.

5. Did the data change in any way across the multiple sessions of recordings? Did the results about coverage, directionality and concentration of peak locations in the open area change across the multiple sessions?

We thank the question from the reviewer. We separated the data per animal and corroborated that there are no outliers or that the observed significances are due to individual animals. We checked the distribution of sparsity (coverage), spatial information, coherence, directional correlation of neuronal activity and the distributions of neuronal peak firing locations for the CC and CO configurations. No differences arose between different animals, except for the correlation values for the CC configuration in rat 3, which were significantly lower than those of rat 2 and rat 5. However, after removing the data from rat 3 from the data set, the observations were still maintained.

We have added a couple of lines about this control in the statistics (methods section).

6. Please show the main effects separated by animal sex

We acknowledge that this is something that should be addressed. Unfortunately, and restricted by our animal licence, we have only worked with male rats. Nevertheless, it is an approach we will include in future research projects in the lab. We thank the reviewer for creating awareness of this necessity in present-time research.

7. Provide examples of real histology photos instead of diagrams

We thank the reviewer for the comment. We have added histological images in Figure 1 —figure supplement 1.

8. Add experimental details showing for each animal how many sessions in ELM were obtained, how many trials and across how many days

We thank the reviewer for the suggestion, which helps to grasp better the behavioural protocol performed. We have added these numbers in the methods section (page 16, lines 473-475 and 491 -503; page 17 lines 504-506).

9. I am having trouble understanding what is plotted in Figure 5c. The legend says "Box plots show median, 25th and 75th percentile". But it is the median of what? What is plotted? Is it possible to plot these data in some other way?

We apologise for the confusion generated by our lack of clarity in the description. We have improved the description in the results and figure legends. In summary, the boxplots are the performance values of sessions where we have been shuffling the trial IDs (proximal and distal exploration). These are our shuffled distributions. However, to have all the sessions included in the same plot, we have subtracted the performance of the actual observed data from the shuffled ones. Therefore, the 0 values are the observed optimal performances, and the boxplots tend to have a negative value because the shuffling performance is reduced (Shuffling perf. – Observed perf.).

[Editors’ note: further revisions were suggested prior to acceptance, as described below.]

Reviewer #1 (Recommendations for the authors):The authors have addressed most of my concerns. However, there are still some concerns that may need to be addressed (see below).– (Figure 3A) The authors nicely illustrated single-cell examples in Figure 3 (and figure supplement 1A and 1B) to show broader place fields in the ventral hippocampus. However, when examining the individual cases, I am concerned that the z-transformed population rate maps in Figure 3A may give the reader the wrong impression that most cells in the ventral hippocampus have focal place fields. Furthermore, inhibitory interneurons in the hippocampus also have their preferred firing locations where their firing rates are higher than others (Ego-Stengel and Wilson, 2007, Hippocampus). As the current study didn't differentially analyze putative pyramidal cells and interneurons, it might be difficult to distinguish between activities in pyramidal neurons and interneurons when using the z-transformed rate maps. Therefore, it seems inappropriate to show the population activities using a z-transformed population rate map.

We followed the advice of the referee and re-plotted the same data also without the z-scoring (new Figure 3- supplement 1C). In addition, we have added cautionary sentences into the result section, as well as to the figure legend :”It is important to note that most of the neurons do not have a spatially restricted activity similar to a typical place cell in the dorsal CA1 hippocampus, but still they exhibit a peak of activity associated to a spatial bin” and “Note that firing rates of neurons remain mostly in the middle ranges (green colour), which is different from classical place cells of the dorsal hippocampus.”

– (Figure 6 —figure supplement 1B) The authors used the averaged speed in the arena spanning 15 cm before crossing the open area to argue that there was no behavioral difference between proximal and distal exploration. However, 15 cm is an arbitrarily determined value, and I wonder if it can serve as a representative measurement to compare animal behavior between proximal and distal trials. This is mainly because they calculated firing rates for SVM analysis from the moment heading towards the open area, not the 15cm before crossing the boundary between closed and open areas. Thus, it might be much more appropriate to compare the speed from the moment heading toward the open area to the end of the closed area.– Additionally, the authors should include statistical testing in Figure 6 —figure supplement 1B. Based on my observations, the speed of future proximal exploration seems significantly lower than that of future distal exploration in session 5. Along with the results of each session, it might be necessary to pool the data from all sessions and perform statistical testing in order to argue that there was no speed difference between proximal and distal exploration.

We thank the reviewer for the comments. We have changed the speed boxplots to the same period used to calculate the firing rate. In addition, we have added the p-values in the figure for each session (only one of them shows significantly different speeds between proximal and distal exploration). We have also pooled all trials for all the animals and added the boxplot in the figure. We labelled it as ‘All Sessions’. We also did the analyses pooled per animal but found no significant differences; therefore, we are not showing those boxplots.

– (Discussion) Keinath and colleagues (Hippocampus, 2014) argued that cells in the mouse ventral hippocampus showed more spatially selective firing patterns when aversive odor (i.e., predator's urine) was introduced in the open arena. This result contradicts the current manuscript because adding an anxiety factor seemed to decrease spatial firing characteristics in the current study as opposed to the results of Keinath et al. (2014). If the authors explain the potential factors of why there was a contradiction between Keinath and colleagues and the current study, it will be helpful to understand the importance of the ventral hippocampus in emotional information processing. Additionally, if the authors explain why the firing fields became larger but not smaller in anxiogenic space in processing emotional processing, it can give readers a clue about how the ventral hippocampus is involved in processing emotional information.

We thank the reviewer for pointing the Keinath et al. (Hippocampus, 2014). We have included the paper in the discussion. Interestingly, we observed similar results in some aspects. We also observe remapping of neuronal activity in aversive conditions and, in fact, thanks to the suggestion of the reviewer, a tendency, yet no significance, of the neurons to increase their firing rate in the CO configuration compared to the CC (Figure 4 —figure supplement 1C). However, in our study, the activity of the neurons in vH increases its coverage, while in Keinath et al., it is reduced (smaller place fields). This type of conflicting finding is also found in dorsal hippocampus studies (we mentioned in the discussion Moita et al. and Scheutte et al.), and the origin of the differences might be attributed to different causes. The most notable difference between our study and Keinath et al. is that for them, the aversive stimulus is presented in a small area (4 squared cm in comparison with the arena area, which is close to 962 squared cm) while in our case the source of the anxiogenic stimulus cannot be pinpointed to a specific part of the open arm, most likely eliciting a less spatial related activity. In Keinath et al., the effects in place field size of vH neurons in-between conditions are not statistically marked in the barplot of figure 3H but in a report in table 1. Still, no post hoc statistical validation was performed, leaving us with an overall significant interaction between the condition and the region, but not a statistical comparison between neutral and aversive odour in the place field size of ventral hippocampal neurons. Nevertheless, these types of minor discrepancies are a motivation to study in even more detail the spatial vs emotional processing in the vH.

Reviewer #3 (Recommendations for the authors):If the authors used only male mice this fact must be stated in the abstract.

We thank the reviewer and we have now added the use of male rats in the abstract.